# Interruption of post-Golgi STING trafficking activates tonic interferon signaling

Xintao Tu [1], Ting-Ting Chu[1], Devon Jeltema [1], Kennady Abbott [1], Kun Yang [1], Cong Xing [1], Jie Han[1], Nicole Dobbs[1] & Nan Yan [1,2] ✉

Activation of the cGAS-STING pathway is traditionally considered a "trigger-release" mechanism where detection of microbial DNA or cyclic di-nucleotides sets off the type I interferon response. Whether this pathway can be activated without pathogenic ligand exposure is less well understood. Here we show that loss of Golgi-to-lysosome STING cofactors, but not ER-to-Golgi cofactors, selectively activates tonic interferon signalling. Impairment of post-Golgi trafficking extends STING Golgi-dwell time, resulting in elevated immune signalling and protection against infection. Mechanistically, trans-Golgi coiled coil protein GCC2 and several RAB GTPases act as key regulators of STING post-Golgi trafficking. Genomic deletion of these factors potently activates cGAS-STING signalling without instigating any pathogenic trigger for cGAS. *Gcc2*[−/−] mice develop STING-dependent serologic autoimmunity. *Gcc2*-deleted or *Rab14*-deleted cancer cells induce T-cell and IFN-dependent anti-tumour immunity and inhibit tumour growth in mice. In summary, we present a "basal flux" mechanism for tonic cGAS-STING signalling, regulated at the level of post-Golgi STING trafficking, which could be exploited for cancer immunotherapy.

The cGAS-STING pathway plays an important role in infection, autoimmunity, cancer and neurodegenerative diseases. Mammalian cGAS and STING are usually activated by microbial DNA or cyclic di-nucleotides, respectively, to induce type I interferon (IFN) signaling. Damage to host organelles such as the nucleus (or nuclear DNA damage) or the mitochondrion also release self-genomic DNA (gDNA) or self-mitochondrial DNA (mtDNA), respectively, that activates cGAS-STING signaling[1]. Until recently, cGAS and STING were considered to be inactive at homeostasis; only exposure to exogenous or endogenous ligands would activate this pathway. However, this 'trigger-release' mechanism does not explain several physiological observations. First, *cGas*[−/−] and *Sting*[−/−] cells and mice are more susceptible to a wide range of RNA virus infections and yet only a small number of these RNA viruses cause pathogenic release of mtDNA that activates cGAS[2–4]. Second, gain-of-function SAVI mutants (STING-associated vasculopathy with onset in infancy) and loss-of-function COPA mutants chronically activate STING signaling via trafficking alterations without

evidence of incurring pathogenic self-DNA exposure[5–7]. Therefore, a broader understanding of cGAS-STING signaling at homeostasis is needed.

STING activation requires trafficking from the ER to ER-Golgi intermediate compartment (ERGIC) and the Golgi, which is essential for recruitment and activation of kinase TBK1 and transcription factor IRF3 that induces IFN expression[8,9]. STING continues trafficking to post-Golgi vesicles and lysosomes, leading to rapid degradation of STING protein and dampening of IFN signaling[10]. Very little is known about the post-Golgi trafficking phase of STING biology. We recently performed a spatiotemporal-resolved proximity labelling screen that defined STING interactomes on various organelles along its trafficking route and characterized one lysosomal cofactor, NPC1[11].

In the current study, we comprehensively analyze STING cofactors from all major organelles during trafficking for their function on the cGAS-STING pathway. We find that STING continuously moves from the ER to lysosomes at homeostasis. Interruption of STING post-

[1]Department of Immunology, UT Southwestern Medical Center, Dallas, TX, USA. [2]Department of Microbiology, UT Southwestern Medical Center, Dallas, TX, USA. ✉e-mail: nan.yan@utsouthwestern.edu

Golgi trafficking, either by genetic knockout of trafficking cofactors or by temperature shift, prolongs STING Golgi-dwell time and enhances IFN signaling. These findings reveal the dynamic nature of STING trafficking at homeostasis as well as an alternative strategy to activate STING signaling.

## Results

### Knockdown of STING post-Golgi cofactors activates tonic IFN signaling

To visualize top hits from the STING trafficking cofactor screen[11], we created a spatial map of candidate cofactors that were captured at various times during STING trafficking through the secretory pathway

(Fig. 1a). We next performed siRNA knockdown to functionally validate 31 selected candidates (based on their ranking in the primary screen and subcellular localization) in two assays: ligand activation and tonic activation. In the ligand activation assay, cells were stimulated with dsDNA that activates cGAS, which produces cGAMP that activates STING trafficking and downstream *Ifnb1* mRNA expression (Fig. 1b and Supplementary Fig. 1a). Knockdown of ER exit site protein SEC24C and ER protein VAPB substantially decreased ligand-mediated IFN response. In contrast, knockdown of several Golgi- and post-Golgi vesicle trafficking proteins, such as Golgi protein RAB6B and trans-Golgi network (TGN) protein GCC2, substantially increased ligand-mediated IFN response. These findings are consistent with ER-exit

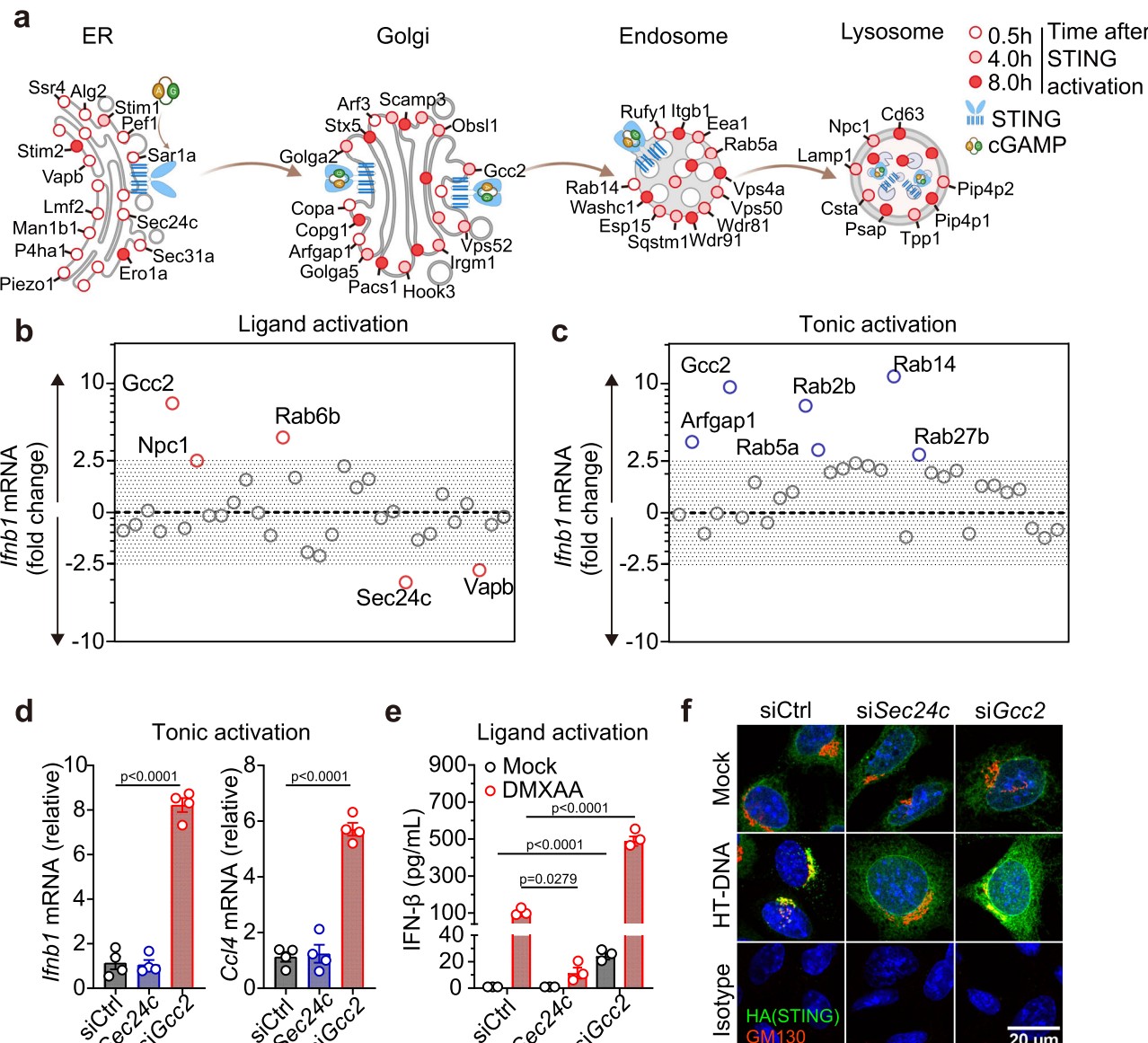

**Fig. 1 | Knockdown of STING post-Golgi cofactors activates tonic IFN signaling. a** A spatiotemporal map of selected STING cofactor candidates identified in the primary proteomic screen. Open, half-filled and filled circles indicate cofactors identified at 0.5 h, 4 h and 8 h after STING activation, respectively. **b** *Ifnb1* mRNA expression in ligand activation assay and tonic activation assay **c**. Wild-type MEFs were transfected with specific siRNA against each of the 31 selected candidate cofactors. Then, knockdown cells were either simulated with HT-DNA (1 μg/mL, 4 h) then qRT-PCR for *Ifnb1* expression **b** or directly measured *Ifnb1* expression without stimulation **c**. Fold-changes were determined by normalizing to control siRNA in either assay. Primary data are shown in Supplementary Fig. 1. **d** qRT-PCR analysis of resting-state *Ifnb1* and *Ccl4* (an ISG) mRNA expression after control, *Sec24c* or *Gcc2* siRNA knockdown in wild-type MEFs. *n* = 4. **e** ELISA analysis of mouse IFN-β in the supernatant after control, *Sec24c* or *Gcc2* siRNA knockdown followed by DXMAA stimulation (50 μM, 8 h). *n* = 3. **f** Confocal microscopy images of STING colocalization with Golgi. *Sting*[KO] MEFs stably expressing HA-STING were transfected with control, *Sec24c* or *Gcc2* siRNA followed by mock or HTDNA stimulation (1 μg/mL, 1.5 h). HA-STING in green, GM130 (a Golgi marker) in red and DAPI in blue. Scale bar, 20 μm. Data are representative of at least three independent experiments. Data (**d** and **e**) are shown as mean ± s.e.m. *P* values were determined by One-way ANOVA.

being critically important for STING activation[8,9,12] and post-Golgi trafficking of STING dampening signaling[10,11].

In the tonic activation assay, we directly measured resting state *Ifnb1* mRNA expression after each cofactor knockdown (Fig. 1c and Supplementary Fig. 1b). Surprisingly, knockdown of Gcc2 and several post-Golgi Rab GTPases (Rab14, Rab2b, Rab5a, Rab27b) significantly increased *Ifnb1* mRNA expression at the resting state (Fig. 1c). Knockdown of Arfgap1, which is required for formation of COPI vesicles and Golgi-to-ER retrograde trafficking[13], also activated tonic IFN signaling, similar to that observed in COPA syndrome[6,7]. These data suggest that selective interruption of Golgi-exit or post-Golgi vesicle trafficking unexpectedly activates innate immune signaling.

We further compared a representative ER cofactor SEC24C and post-Golgi cofactor GCC2. SEC24C is known to regulate STING ER-exit[12]. GCC2 is a coiled-coil protein involved in post-Golgi vesicle sorting that has not been implicated in STING signaling before. *Gcc2* knockdown in wild-type MEFs significantly increased baseline *Ifnb1* and *Ccl4* (an IFN-stimulated gene, ISG) mRNA expression compared to control knockdown, whereas *Sec24c* knockdown had no effect (Fig. 1d and Supplementary Fig. 1c). Upon stimulation with the STING agonist DMXAA, *Sec24c* knockdown significantly decreased while *Gcc2* knockdown increased IFN-β production (Fig. 1e). Confocal microscopy further revealed that DNA stimulation led to rapid STING trafficking from the ER to post-Golgi vesicles in control knockdown cells. *Sec24c* knockdown impaired STING ER-exit and STING remained largely on the ER after DNA stimulation. *Gcc2* knockdown impaired Golgi-exit, leading to STING accumulation on the ER and the Golgi after DNA stimulation (Fig. 1f). Collectively, these data suggest that STING Golgi-exit is regulated by GCC2 and other post-Golgi co-factors that are essential for attenuating STING signaling.

## GCC2 is required for STING Golgi-exit

GCC2 belongs to a family of 11 coiled-coil proteins called Golgins that direct cargo trafficking in and out of the TGN. Four Golgins (GCC1, GCC2, GOLGA1, and GOLGA4) are located on the TGN, which direct capture or release of cargos along the secretary pathway (Fig. 2a)[14]. Redundancy amongst these Golgins ensures that knocking down any one Golgin would not cause a global 'traffic jam' in the secretory pathway[15]. We knocked down each Golgin individually in wild-type MEFs, and surprisingly only *Gcc2* knockdown significantly increased tonic as well as ligand-stimulated IFN response (Fig. 2b and Supplementary Fig. 1d). This data suggests that STING Golgi-exit specifically requires GCC2. We next generated *Gcc2^{KO}* MEFs by CRISPR/Cas9. *Gcc2^{KO}* MEFs showed increased baseline expression of IFN genes and ISGs compared to *Gcc2^{WT}* MEFs, which was abolished when *Sting*, but not *Mavs*, was further knocked down (Fig. 2c and Supplementary Fig. 1e, confirmation with double knockout cells below).

Western blot analysis of STING signaling kinetics revealed much more sustained levels of p-Sting, p-Tbk1 and p-Irf3 in *Gcc2^{KO}* cells compared to *Gcc2^{WT}* cells (Fig. 2d). Increased STING protein and p-Tbk1 were also detectable in unstimulated *Gcc2^{KO}* cells, consistent with tonic IFN activation. *Sting* mRNA was slightly increased in *Gcc2^{KO}* cells compared to *Gcc2^{WT}* (Supplementary Fig. 1f). STING protein is degraded by the lysosome at homeostasis as well as after ligand activation[10,16]. We found that STING protein degradation was also impaired in *Gcc2^{KO}* cells, consistent with interrupted post-Golgi trafficking to the lysosome (Fig. 2d). To substantiate this, we treated *Gcc2^{WT}* and *Gcc2^{KO}* cells with Bafilomycin A1 (BafA1) that inhibits lysosome function. BafA1 treatment impaired HT-DNA stimulated STING degradation in *Gcc2^{WT}* cells as expected. In *Gcc2^{KO}* cells, STING degradation is already severely delayed and BafA1 treatment did not cause further delay (Supplementary Fig. 2a). We also measured other signaling activities of STING; NF-kB was not affected while autophagy was slightly altered, suggesting that not all STING activities are affected by post-Golgi trafficking interruption (Supplementary Fig. 2b). Cell

surface IFNAR1 expression and total IFNAR1 protein levels were similar in *Gcc2^{WT}* and *Gcc2^{KO}* cells (Supplementary Fig. 2c).

We next infected *Gcc2^{WT}* and *Gcc2^{KO}* MEFs with HSV-1-GFP. Compared to *Gcc2^{WT}*, *Gcc2^{KO}* cells showed higher levels of IFNβ production (Fig. 2e) and reduced viral replication and viral titer (Fig. 2e, f). We also knocked down *GCC2* in human THP-1 cells and HeLa cells and observed increased *IFNB1* and *CXCL10* mRNA expression (Supplementary Fig. 2d). These data confirm that GCC2 is a critical negative regulator of STING signaling by mediating its trafficking from the Golgi to the lysosome for degradation both at the resting-state and after ligand stimulation.

We next used quantitative time-lapse confocal microscopy to analyze endogenous STING trafficking and signaling activation. We also defined key trafficking parameters such as ER-exit, Golgi-exit and Golgi-dwell time using three methods: (1) By examining endogenous STING colocalization with organelle markers (Supplementary Figs. 3a, b and 4a, b). ER-exit is defined by decreased STING colocalization with the ER marker PDI and concomitant increased colocalization with the cis-Golgi marker GM130 or trans-Golgi marker TGN38. Golgi-exit is defined by loss of colocalization with Golgi markers. (2) By measuring STING protein density (Supplementary Fig. 3c). Endogenous STING protein density is low at homeostasis when spreading out on the ER. STING protein condenses drastically (by oligomerization) after translocation to the Golgi, and then STING protein density quickly decreases after dispersing into post-Golgi vesicles and eventual degradation by the lysosome. (3) Live-cell microscopy of STING-GFP (Supplementary Fig. 3d). The STING protein density assay also allows live-cell microscopy analysis of STING-GFP in MEFs. We calculated the time from ER-exit (turn-on signal) to Golgi-exit (turn-off signal) as 'Golgi-dwell' time. All three measurements of Golgi-dwell time yielded similar results (Supplementary Fig. 3b–d).

We predicted that a longer STING Golgi-dwell time could allow more recruitment and phosphorylation of TBK1 and IRF3, thus enhancing IFN signaling. We then analyzed *Gcc2^{WT}* and *Gcc2^{KO}* cells. STING ER-exit time was slightly delayed, but STING Golgi-exit was substantially delayed in *Gcc2^{KO}* cells, leading to significantly increased Golgi-dwell time compared to *Gcc2^{WT}* cells (Fig. 2g). We observed similar increased Golgi-dwell time using either cis-Golgi or trans-Golgi marker for STING colocalization (Supplementary Fig. 4a–d). We also measured key biochemical markers of STING activation (p-TBK1 and p-IRF3) using quantitative microscopy. We detected higher p-Tbk1 and p-Irf3 signal activity in *Gcc2^{KO}* compared to *Gcc2^{WT}* cells, consistent with increased IFN response (Fig. 2h, i, and Supplementary Fig. 3e, f). Importantly, the stacking and morphology of the Golgi remain normal in *Gcc2^{KO}* cells and are indistinguishable to *Gcc2^{WT}* cells, indicating that *Gcc2^{KO}* does not cause disruption of Golgi structure in MEFs (Fig. 2g and Supplementary Fig. 3g, h). These data suggest that GCC2 regulates STING Golgi-exit and *Gcc2^{KO}* enhances both tonic and ligand-mediated STING signaling.

## Temperature-mediated slowdown of vesicle trafficking enhances STING signaling

We next evaluated whether slowdown of vesicle trafficking via thermoregulation also enhances STING signaling. Lowering cell culture temperature from 37 °C to 20 °C is known to slowdown vesicle trafficking at the Golgi[17] (Fig. 3a). Remarkably, cells cultured at 20 °C showed drastically increased and more sustained STING signaling activities as indicated by prolonged phosphorylation of Sting, Tbk1 and Irf3 compared to cells cultured at 37 °C (Fig. 3b). Quantitative microscopy analysis revealed remarkably extended STING Golgi-dwell time and increased p-Tbk1 in cells at 20 °C compared to 37 °C (Fig. 3c, d). To control for possible differences between transfection efficiency at 37 °C and 20 °C, we also stimulated cells with a cell-permeable STING agonist DMXAA. Similar to HT-DNA, DMXAA stimulated more sustained STING signaling activation at 20 °C compared to 37 °C

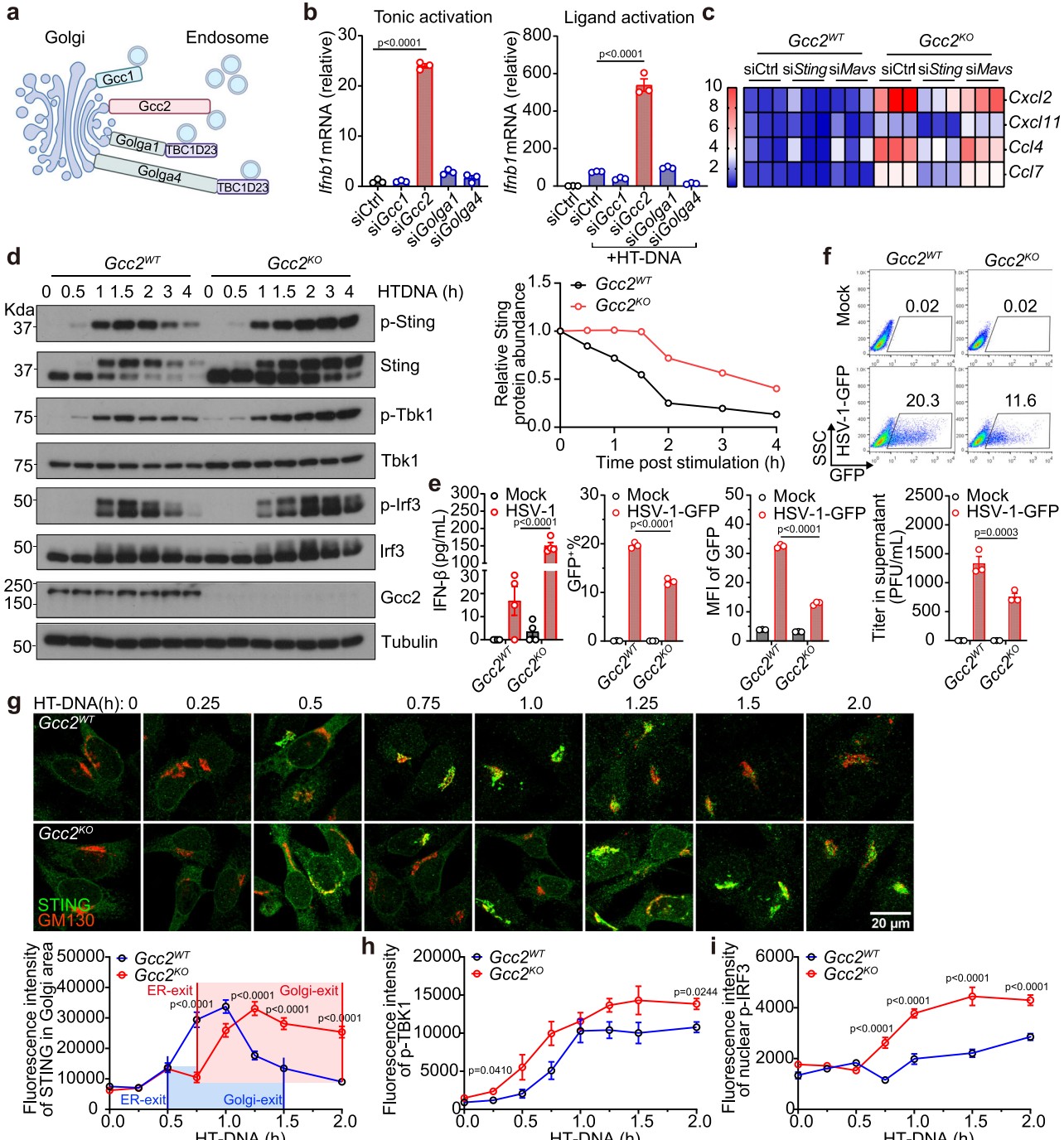

**Fig. 2 | GCC2 is required for STING Golgi-exit. a** A diagram of Golgin family members at the trans-Golgi network (TGN). **b** qRT-PCR analysis of tonic (left) or ligand-activated (right) *lfnb1* mRNA expression after mock knockdown (siCtrl) or specific siRNA knockdown of *Gcc1*, *Gcc2*, *Golga1*, or *Golga4* in wild-type MEFs. $n = 3$. **c** A heatmap showing expression of indicated ISGs (right) at the resting-state in *Gcc2^WT* and *Gcc2^KO* MEFs after control, *Sting*, *Mavs* siRNA knockdown. **d** Western blot analysis of STING signaling kinetics. *Gcc2^WT* and *Gcc2^KO* MEFs were stimulated with HTDNA (1 μg/mL) for indicated times (top). Quantification of relative Sting protein abundance (normalized to Tubulin, then set 0 h value to 1) are shown on the right. p-Sting, p-Tbk1, and p-Irf3 are key phosphorylation events of the STING signaling pathway. **e** ELISA analysis of mIFN-β in the supernatant in *Gcc2^WT* and *Gcc2^KO* MEFs 24 h after HSV-GFP infection (m.o.i. = 1). $n = 4$. **f** HSV-1-GFP infection in *Gcc2^WT* and *Gcc2^KO* MEFs. Top, representative FACS plot of GFP expression 24 h after HSV-1-GFP infection (m.o.i. = 1). Lower panels, percentage of GFP+ cells (lower left), GFP MFI (Mean fluorescence intensity, lower middle) and HSV-1 titers in the supernatant (lower right) in *Gcc2^WT* and *Gcc2^KO* MEFs. **g**–**i** Confocal microscopy analysis of endogenous STING trafficking and signaling. *Gcc2^WT* or *Gcc2^KO* MEFs were stimulated with HT-DNA (1 μg/mL) for indicated times (top). Cells were then stained for endogenous STING (green) and Golgi marker GM130 (red) (**g**, top). Quantification of fluorescence intensity of endogenous STING in the Golgi (GM130+ area) using Fiji (**g**, bottom). A similar set of cells were stained for p-Tbk1 and p-Irf3 (images in Supplementary Fig. 3e, f). Quantification of fluorescence intensity of endogenous p-TBK1 (**h**) and p-IRF3 (**i**) in *Gcc2^WT* and *Gcc2^KO* MEFs are shown. The colored area corresponds to Golgi-dwell time. $n = 3$. Data are representative of at least three independent experiments. At least 14 cells in two different views were analyzed (**g**–**i**). Data (**b**, **e** and **g**–**i**) are shown as mean ± s.e.m. *P* values were determined by one-way ANOVA (**b** and **e**) or by two-way ANOVA (**g**–**i**).

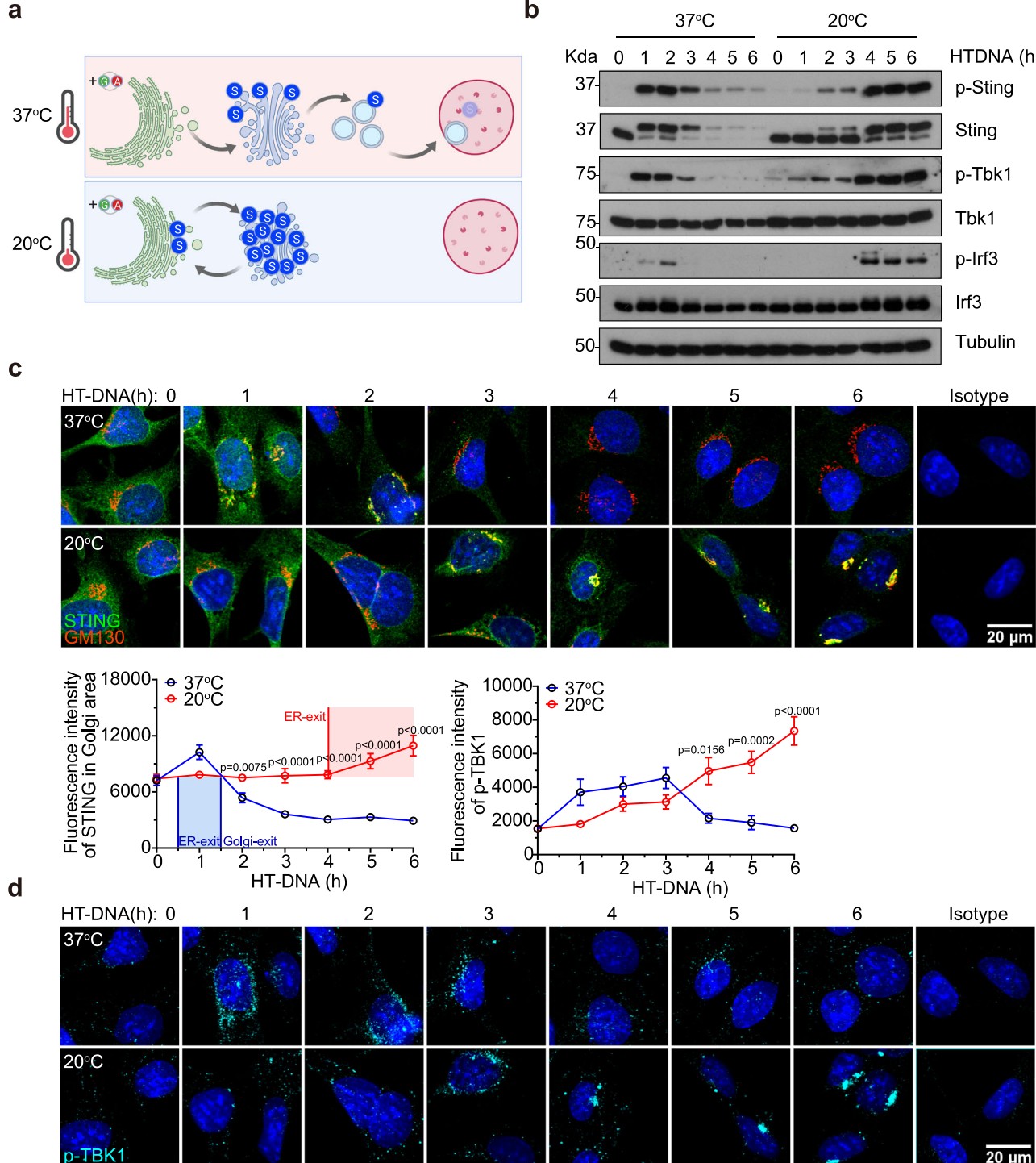

**Fig. 3 | Temperature-mediated slowdown of trafficking at the Golgi enhances STING signaling. a** A diagram of expected STING trafficking at 37 °C and 20 °C. Top, at normal cell culture temperature (37 °C), STING activation triggers trafficking from the ER to Golgi then to endolysosomes. Bottom, lowering the temperature to 20 °C slows down post-Golgi vesicle trafficking, leading to STING accumulation on the Golgi. **b** Western blot analysis of STING signaling kinetics. Wild-type MEFs were first acclimated in 37 °C or 20 °C then stimulated with HT-DNA (1 μg/mL) for indicated times (top) while remaining at the same temperature. Control experiments with DMXAA or poly(I:C) stimulation are shown in Supplementary

Fig. 5. **c** and **d** Confocal microscopy analysis of endogenous STING trafficking and signaling. Wild-type MEFs were stimulated with HT-DNA (1 μg/mL) while incubated at 37 °C or 20 °C. Endogenous STING trafficking (**c**, images on top, quantifications at the bottom) and signaling activation (**d**, p-TBK1 quantification on top, images at the bottom) were analyzed as in Fig. 2g, h. Scale bar, 20 μm. $n = 3$. Data are representative of at least three independent experiments. At least 17 cells in two different views were analyzed (**c**, **d**). Data (**c**, **d**) are shown as mean ± s.e.m. *P* values were determined by Two-way ANOVA (**c**, **d**).

(Supplementary Fig. 5a). Further, polyI:C transfection (that activates a parallel RNA sensing pathway) stimulated similar kinetics of pTbk1 and pIrf3 at either temperature (Supplementary Fig. 5b). These data suggest that STING signaling is selectively affected by post-Golgi trafficking slowdown and further support that Golgi-exit is a critical checkpoint for attenuating STING signaling. Extending STING Golgi-dwell time by selective interruption (via $Gcc2^{KO}$) or slowdown (via lowing temperature to 20 °C) of post-Golgi vesicle trafficking substantially enhances IFN signaling capacity.

## cGAS drives tonic STING trafficking and signaling

We next asked what drives STING trafficking and signaling in $Gcc2^{KO}$ cells. We first compared $Gcc2^{WT}$, $Gcc2^{KO}$, $Gcc2^{KO}cGas^{KO}$ and $Gcc2^{KO}Sting^{KO}$ cells. Western blot analysis of biochemical features of the cGAS-STING pathway showed strong activation of p-Sting, p-Tbk1, p-Stat1 in $Gcc2^{KO}$ but not $Gcc2^{WT}$, $Gcc2^{KO}cGas^{KO}$ and $Gcc2^{KO}Sting^{KO}$ cells. ISG expression in $Gcc2^{KO}$ cells were also abolished in $Gcc2^{KO}cGas^{KO}$ and $Gcc2^{KO}Sting^{KO}$ cells (Fig. 4a, b). These data suggest that innate immune activation in $Gcc2^{KO}$ cells requires both cGAS and STING.

We next investigated molecular functions of cGAS and STING. We reconstituted $Gcc2^{KO}cGas^{KO}$ cells with wild-type cGas, E225A/D227A mutant (disrupts enzymatic activity) or K407A/K411A (disrupts DNA binding)[18]. Only wild-type cGAS restored STING signaling, suggesting both DNA binding and enzymatic activity of cGAS are required for tonic innate immune activation in Gcc2-KO cells (Fig. 4c, d). We also reconstituted $Gcc2^{KO}Sting^{KO}$ cells with wild-type Sting or R238A/Y240A mutant (disrupts cGAMP binding) or S366A (disrupts STING phosphorylation)[12,19]. Only wild-type Sting restored IFN signaling, suggesting both cGAMP binding and STING phosphorylation by TBK1 are essential (Fig. 4e, f).

The sources of DNA driving tonic cGAS activity in wild-type cells has been defined in a previous study[20]. We performed similar endogenous cGAS immunoprecipitation experiments and quantified cGAS-bound DNA substrates. We found that resting state cGAS bound to satellite DNA, mitochondria DNA, and retroelement DNA but not genomic DNA (Supplementary Fig. 6a). We next explored possibilities of mitochondrial damage or nuclear DNA damage that would release pathogenic self-DNA to activate cGAS. We treated $Gcc2^{KO}$ cells with EtBr to deplete mtDNA, which did not reduce ISG expression (Fig. 4g, h). In a control experiment, ABT-737+QvD-Oph treatment (that is known to cause mtDNA release) induced $Ifnb1$ and $Cxcl10$ mRNA expression, which was abolished after mtDNA depletion by EtBr (Supplementary Fig. 6b). Confocal microscopy analysis of mitochondrial morphology also did not reveal any abnormality in $Gcc2^{KO}$ cells (Supplementary Fig. 6c). DNA damage marker, p-ATM, did not change between $Gcc2^{WT}$ and $Gcc2^{KO}$ cells (Supplementary Fig. 6d). Further, intracellular cGAMP levels were similar in both $Gcc2^{WT}$ and $Gcc2^{KO}$ cells (Fig. 4i). We further isolated cytosolic DNA from $Gcc2^{WT}$ and $Gcc2^{KO}$ cells and measured representative categories of known DNA triggers of cGAS, such as retroelements and mtDNA, and we did not detect any difference (Fig. 4j). Collectively, $Gcc2^{KO}$ cells show strong tonic IFN signaling, requirements for both cGAS and STING, but lack evidence of self-DNA accumulation or increased cGAMP production. These data suggest that cGAS-STING is operational at homeostasis; $Gcc2^{KO}$ boosts STING signaling via trafficking interruption without further activating cGAS.

## cGAS maintains the basal state of immune defense

$cGas^{-/-}$ mice are more susceptible to infection of several RNA viruses despite that many RNA viruses do not directly activate mammalian cGAS[2]. We next directly compared wild-type and $cGas^{-/-}$ mouse tissues. Both heart and BMDMs from $cGas^{-/-}$ mice showed significantly lower baseline ISG expression compared to wild-type (Fig. 5a, b). $cGas^{KO}$ MEFs showed a similarly reduced basal state of immune defense compared to wild-type MEFs (Fig. 5c, d). Notably, there are

low but detectable levels of p-Sting, p-Tbk1 and p-Stat in wild-type MEFs, which were further reduced in $cGas^{KO}$ MEFs. Intracellular cGAMP levels are also significantly lower in $cGas^{KO}$ compared to wild-type MEFs, both at homeostasis and after DNA stimulation (Supplementary Fig. 6e).

Reconstitution of $cGas^{KO}$ MEFs with wild-type cGAS, but not DNA-binding or enzymatic mutant cGAS, restored p-Sting, p-Tbk1, p-Stat, and ISG expression (Figs. 5e, f). These data confirm that cGAS is operational at homeostasis and it promotes STING signaling to maintain the basal state of immune defense. This homeostatic activity of cGAS is likely driven by low levels of DNA species that are accessible to cGAS[20].

## RAB14 and other Rabs mediate STING post-Golgi trafficking

Post-Golgi cargos are sorted to various destinations 'postmarked' by distinct Rab GTPases (Rabs)[21]. GCC2 'hands off' cargos to several Rabs[15]. We also identified several Rabs as STING trafficking cofactors from our primary and secondary screens (Fig. 1). To determine if any of these Rabs specifically regulate STING signaling, we individually knocked down a panel of Rabs (that are either known to interact with GCC2 and/or identified in our screen) in wild-type and $Sting^{-/-}$ MEFs. Knockdown of several Rabs individually (Rab2b, Rab6b, Rab9a, and Rab14) significantly increased baseline $Ifnb1$ mRNA expression in wild-type but not in $Sting^{-/-}$ MEFs, suggesting that they specifically act on the STING pathway (Fig. 6a and Supplementary Fig. 7a).

We next performed co-immunoprecipitation (IP) experiments in HEK293T cells. STING interacts strongly with Rab2b, Rab6b, Rab9a, and Rab14, and interacts weakly with Rab23 and Rab27, consistent with their respective effects on STING signaling (Fig. 6b). GCC2 also interacts with Rab2b, Rab6b, and Rab14 but interestingly not with STING (Supplementary Fig. 7b–d). STING interaction with and functional dependence on multiple Rabs suggests that post-Golgi STING vesicles may be more diverse than previously appreciated.

We next focused on Rab14, which is responsible for transporting cargos to the endolysosome[22], where STING is degraded. $Rab14^{KO}$ MEFs show significantly higher baseline expression of several ISGs compared to $Rab14^{WT}$ MEFs and such elevated ISGs were reduced in $Rab14^{KO}Sting^{KO-pool}$ MEFs but not in $Rab14^{KO}Mavs^{KO-pool}$ MEFs (Fig. 6c and Supplementary Fig. 7g). Resting-state STING protein level was substantially higher in $Rab14^{KO}$ compared to $Rab14^{WT}$ MEFs, despite similar $Sting$ mRNA level, consistent with impaired tonic Golgi-to-lysosome STING trafficking (Fig. 6d and Supplementary Fig. 7e, f). Upon ligand stimulation, $Rab14^{KO}$ cells showed increased p-Sting and p-Irf3, as well as delayed STING degradation (Fig. 6d). Quantitative microscopy showed substantially delayed STING Golgi-exit and increased Golgi-dwell time period in $Rab14^{KO}$ compared to $Rab14^{WT}$ MEFs (Fig. 6e). Collectively, $Rab14^{KO}$ phenocopies $Gcc2^{KO}$ and these data suggest that STING Golgi-exit regulated by Gcc2-Rab14 is critical for immune signaling.

## $Gcc2^{-/-}$ mice develop STING-dependent serologic autoimmunity

We next assessed the physiological consequence of interrupting STING post-Golgi trafficking in vivo. STING activation has been associated with autoimmune diseases with a wide range of severity and phenotypes. We generated $Gcc2$ and $Rab14$ deletion alleles by CRISPR/Cas9. $Rab14^{+/-}$ mice were viable but $Rab14^{-/-}$ was embryonic lethal at an early developmental stage, thus precluding any immune pathology analysis. $Gcc2^{-/-}$ mice were born at the expected Mendelian ratio and have normal body weight as wild-type mice (Supplementary Fig. 8a). At 6-month-old, $Gcc2^{-/-}$ mice showed significantly higher levels of IgM autoantibodies against a wide range of antigens commonly associated with lupus and other autoimmune diseases (Fig. 7a, b). Importantly, $Gcc2^{-/-}Sting^{-/-}$ mice completely restored these autoantibodies down to wild-type levels (Fig. 7a, b). IgG

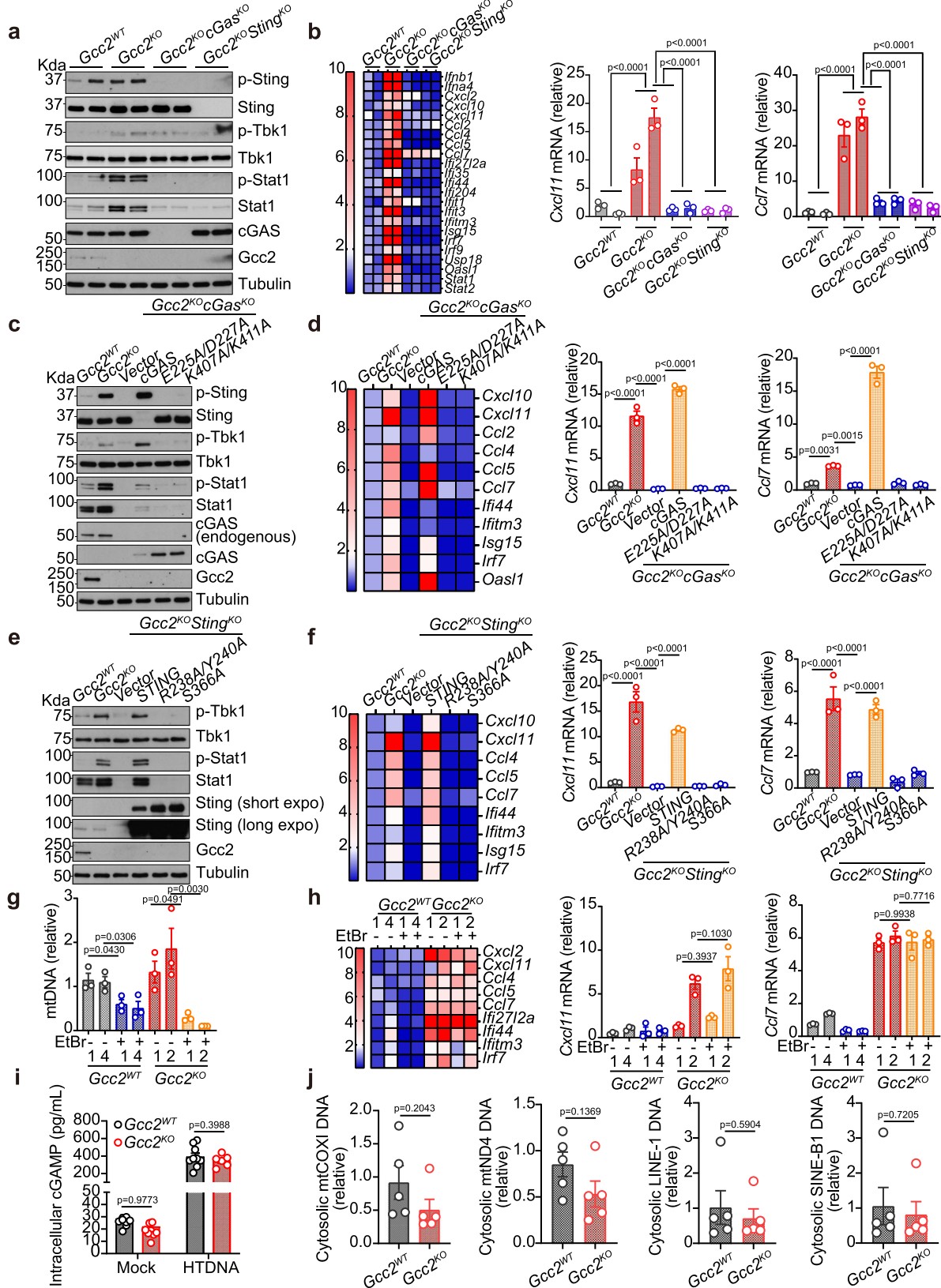

autoantibodies and serum cytokines are normal when comparing *Gcc2⁻/⁻* to wild-type mice (Supplementary Fig. 8a–c). We did not detect any changes in ISG expression in tissues and BMDMs; T-cell and B-cell populations in the peritoneum and spleen are also normal (Supplementary Fig. 9a–c). These data suggest that *Gcc2* deficiency induces STING-dependent autoimmunity in mice.

## *Gcc2ᴷᴼ* and *Rab14ᴷᴼ* induce anti-tumor immunity

STING activation induces anti-tumor immunity mediated by IFN signaling and T cell response[23,24]. We first generated *Sec24c, Gcc2, Rab14* and *Npc1* pooled CRISPR-KO B16 melanoma cells and implanted them in C57BL/6 wild-type mice. All three STING post-Golgi cofactors knockouts (*Gcc2ᴷᴼ⁻ᵖᵒᵒˡ, Rab14ᴷᴼ⁻ᵖᵒᵒˡ, Npc1ᴷᴼ⁻ᵖᵒᵒˡ*) inhibited tumor growth

**Fig. 4 | Both cGAS and STING are required for tonic IFN signaling in Gcc2-KO cells. a, b** Western Blot (**a**) and qRT-PCR (**b**) analysis of tonic IFN signaling in $Gcc2^{WT}$, $Gcc2^{KO}$, $Gcc2^{KO}STING^{KO}$, and $Gcc2^{KO}cGAS^{KO}$ MEFs. Two independent CRISPR/Cas9 knockout clones of each genotype are included. p-Sting, p-Tbk1, and p-Stat1 are key phosphorylation events of the STING signaling pathway (**a**). Expression of IFN genes and ISGs are shown in a heatmap (**b**, left) and representative bar graphs (**b**, right). $n = 3$. **c, d** Western Blot (**c**) and qRT-PCR (**d**) analysis of tonic IFN signaling in $Gcc2^{WT}$, $Gcc2^{KO}$, $Gcc2^{KO}cGAS^{KO}$ MEFs reconstituted with vector control, wild-type cGAS, E225A/D227A cGAS (enzymatic-dead mutant), K407A/K411A cGAS (DNA-binding mutant). Cells were analyzed as in **a, b**. $n = 3$. **e, f** Western Blot (**e**) and qRT-

PCR (**f**) analysis of tonic IFN signaling in $Gcc2^{WT}$, $Gcc2^{KO}$, $Gcc2^{KO}STING^{KO}$ MEFs reconstituted with vector control, wild-type STING, R238A/Y240A STING (ligand-binding mutant), S366A STING (IFN signaling mutant). Cells were analyzed as in **a, b**. $n = 3$. **g, h** qPCR analysis of mtDNA content (**g**) and qRT-PCR analysis of tonic IFN signaling (**h**) in two independent clones of $Gcc2^{WT}$, $Gcc2^{KO}$ with or without mtDNA depletion by EtBr. $n = 3$. **i** Intracellular cGAMP level in $Gcc2^{WT}$ and $Gcc2^{KO}$ cells mock treated or stimulated with HT-DNA. $n = 8$. **j** qPCR analysis of indicated cytoplasmic DNA species in $Gcc2^{WT}$ and $Gcc2^{KO}$ cells. $n = 5$. Data are representative of at least three independent experiments. Data (**b**, **d**, **f**, and **g–j**) are shown as mean ± s.e.m. $P$ values were determined by one-way ANOVA (**b**, **d**, **f**, and **g–j**).

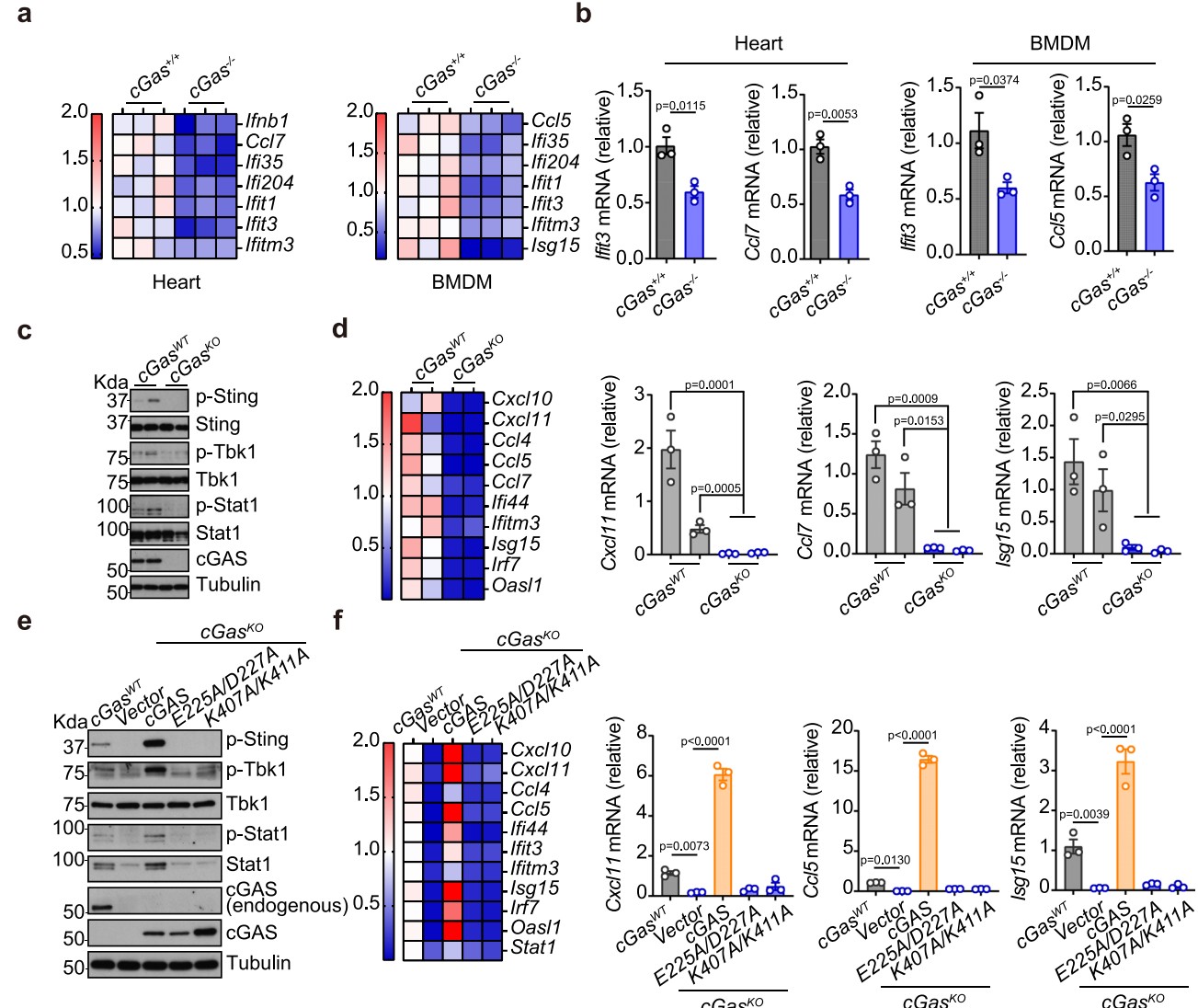

**Fig. 5 | cGAS drives STING signaling to maintain a basal state of immune defense. a, b** qRT-PCR analysis of baseline immune gene expression in $cGAS^{WT}$ and $cGAS^{KO}$ mouse heart or BMDMs ($n = 3$). Heatmaps of multiple ISGs are shown in **a**. Representative bar graphs are shown in **b**. **c, d** Western Blot (**c**) and qRT-PCR (**d**) analysis of tonic IFN signaling in $cGAS^{WT}$ and $cGAS^{KO}$ MEFs. Two independent clones of $cGAS^{WT}$ and $cGAS^{KO}$ MEFs were shown. p-Sting, p-Tbk1 and p-Stat1 are key phosphorylation events of the STING signaling pathway (**c**). Expression of ISGs are

shown in a heatmap (**d**, left) and representative bar graphs (**d**, right). $n = 3$. **e, f** Western blot (**c**) and qRT-PCR (**d**) analysis of tonic IFN signaling in $cGAS^{WT}$ and $cGAS^{KO}$ MEFs reconstituted with vector control, wild-type cGAS, E225A/D227A cGAS (enzymatic-dead mutant), K407A/K411A cGAS (DNA-binding mutant). Cells were analyzed as in **c, d**. $n = 3$. Data are representative of at least three independent experiments. Data (**b**, **d**, and **f**) are shown as mean ± s.e.m. P values were determined by one-way ANOVA (**b**, **d**, and **f**).

(Fig. 8a). Knockout of ER cofactor ($Sec24c^{KO-pool}$) did not inhibit tumor growth (Fig. 8a). We next selected individual clones of $Gcc2^{KO}$ and $Rab14^{KO}$ B16 cells for further analysis. Both $Gcc2^{KO}$ and $Rab14^{KO}$ B16 cells expressed elevated levels of ISG mRNA compared to wild-type

B16 (Supplementary Fig. 10a, b). Wild-type, $Gcc2^{KO}$ and $Rab14^{KO}$ B16 cells proliferated equally in vitro (Fig. 8b). When implanted subcutaneously into immunocompetent wild-type mice, $Gcc2^{KO}$ and $Rab14^{KO}$ B16 tumors grew significantly slower than wild-type B16

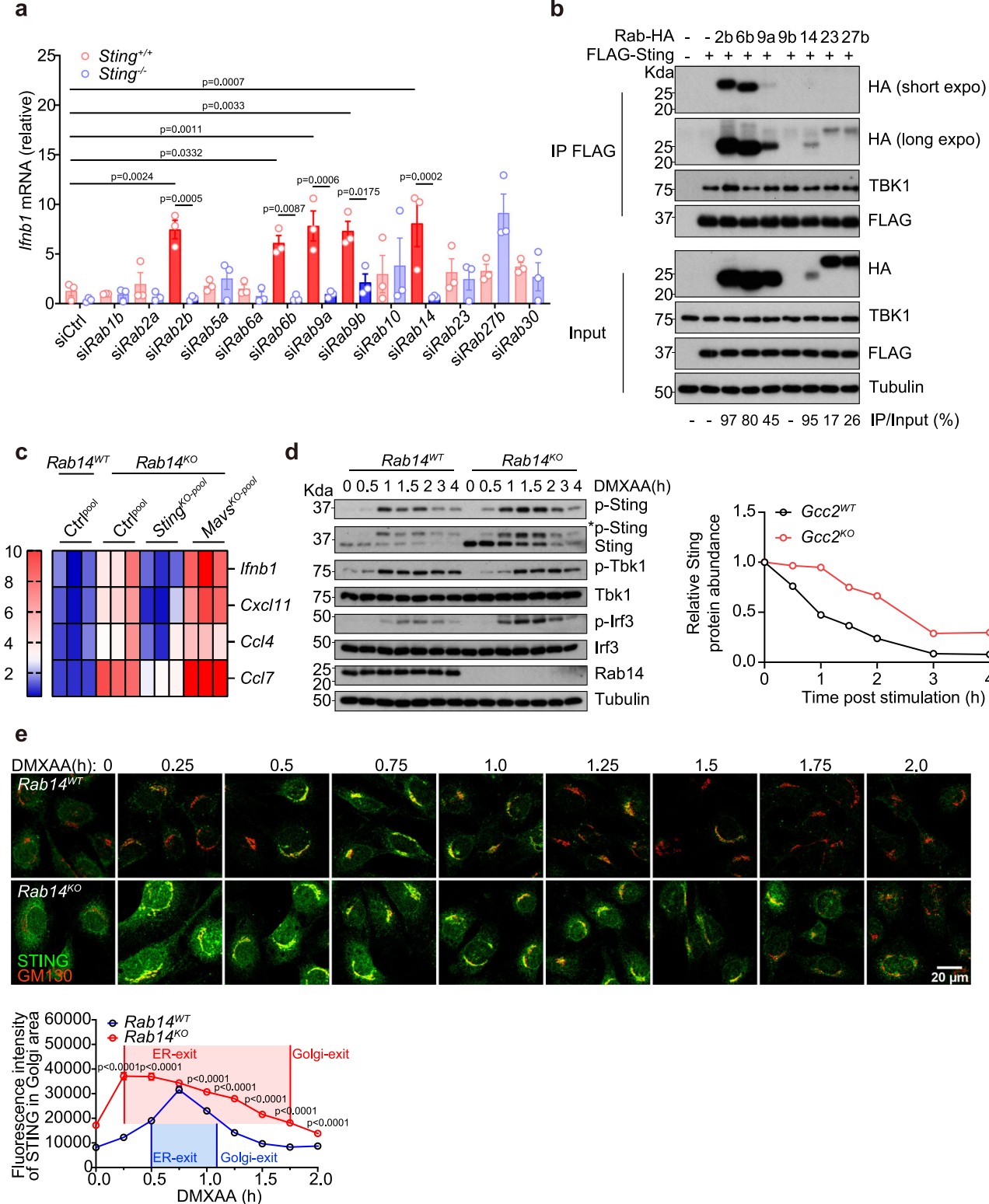

tumors, leading to higher overall survival of tumor-bearing mice (Fig. 8c). To further assess the mechanism of anti-tumor immunity, we measured WT, $Gcc2^{KO}$ and $Rab14^{KO}$ B16 tumor growth in either $Rag1^{-/-}$ or $Ifnar1^{-/-}$ mice. We no longer observe any anti-tumor immunity, suggesting that the anti-tumor response is T-cell mediated and IFN-dependent (Figs. 8d, e). These data suggest that interrupting basal-flux of cGAS-STING at the post-Golgi step can lead to functionally similar anti-tumor immunity as STING agonists in vivo (Fig. 9, a model).

## Discussion

The classical pattern recognition mechanism is central to innate immune receptors and is critical for microbial defense, where the detection of microbial DNA or cyclic di-nucleotides triggers cGAS or STING, respectively, leading to the type I IFN response. This mechanism can be further extended to detection of pathogenic self-DNA, especially in autoimmune disease conditions with clear evidence of defective self-DNA metabolism (e.g. $Trex1^{-/-}$ [25]) or disrupted mitochondrial integrity (e.g. $Tfam^{+/-}$ [26]). Intriguingly, many physiologically

**Fig. 6 | RAB14 and other Rabs mediate STING post-Golgi trafficking. a** qRT-PCR analysis of tonic *Ifnb1* mRNA expression in *Sting^WT* or *Sting^KO* MEFs after siRNA knockdown of indicated Rab GTPases (bottom). Solid red bars indicate siRNAs that induce STING-dependent IFN signaling. *n* = 3. **b** Co-immunoprecipitation analysis of STING:Rab interaction. HEK293T cells were transfected with FLAG-Sting and various HA-Rab GTPases (as indicated on top), IP with the FLAG antibody and blotted with indicated antibody on the right. Percentages of IP/Input are shown on the bottom. Note the expression level of Rab14 plasmid is low in whole cell lysate but IP/Input% is high. **c** A heatmap showing expression of ISGs in *Rab14^WT*, *Rab14^KO*, *Rab14^KO Sting^KO-pool*, and *Rab14^KO Mavs^KO-pool* MEFs. Bar graphs are shown in Supplementary Fig. 7g. **d** Western blot analysis of STING signaling kinetics in *Rab14^WT* and

*Rab14^KO* MEFs. Cells were stimulated with DMXAA (10 μg/mL) for indicated times (top) and blotted for total- and phosphor-proteins as indicated on the right. Quantification of relative Sting protein abundance (normalized to Tubulin, then set 0 h value to 1) are shown on the right. **e** Confocal microscopy analysis of endogenous STING trafficking in *Rab14^WT* and *Rab14^KO* MEFs. Cells were stimulated with DMXAA (10 μg/mL) and analyzed as in Fig. 2g. Endogenous STING in green and GM130 (a Golgi marker) in red. Scale bar, 20 μm. *n* = 3. Data are representative of at least three independent experiments. At least 37 cells in two different views were analyzed (**e**). Data (**a** and **e**) are shown as mean ± s.e.m. *P* values are determined by One-way ANOVA (**a**) or by Two-way ANOVA (**e**).

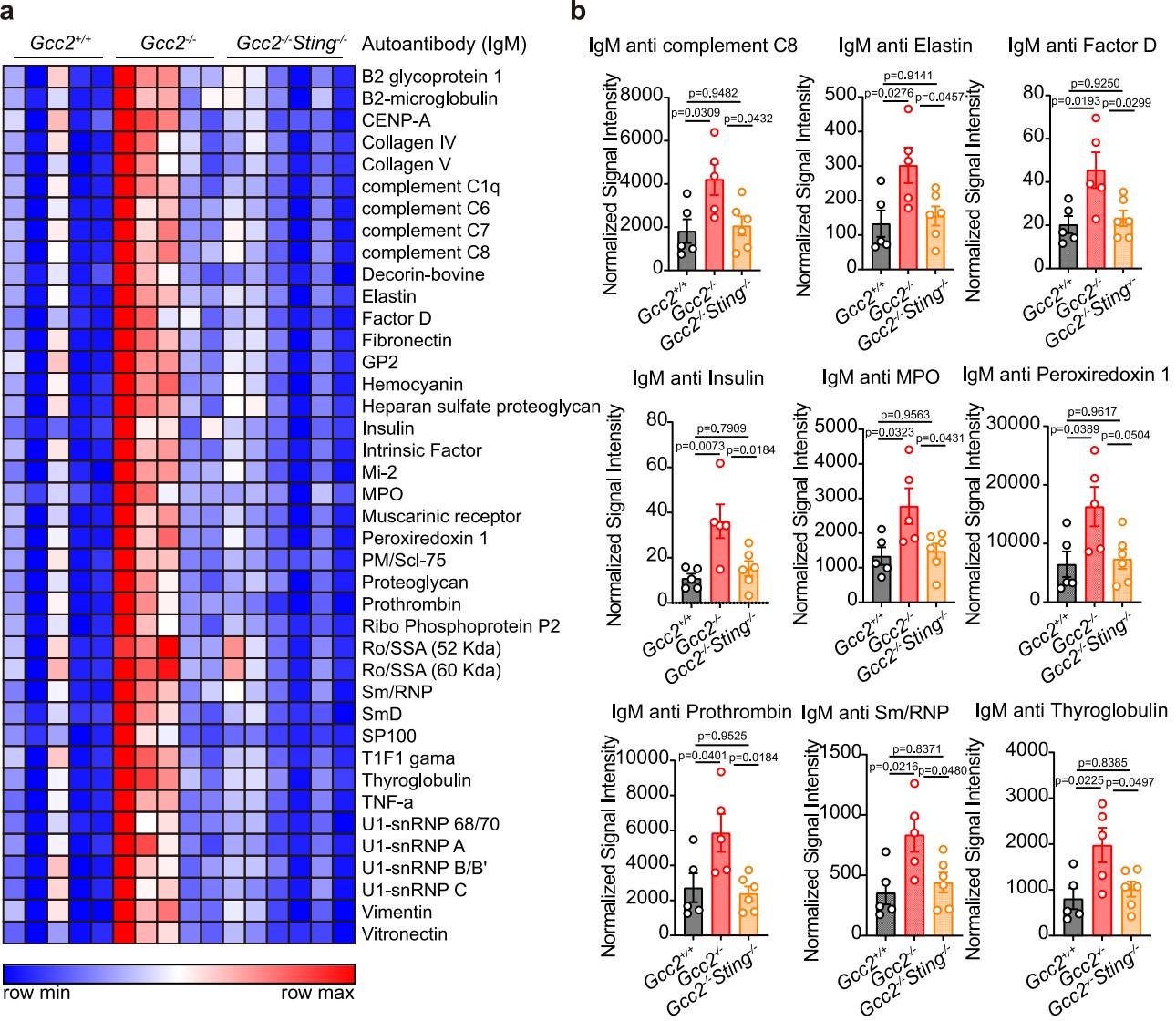

**Fig. 7 | Gcc2^−/− mice develop STING-dependent serologic autoimmunity. a** A heatmap showing IgM autoantibody array analysis of *Gcc2^+/+*, *Gcc2^−/−*, and *Gcc2^−/−Sting^−/−* mouse serum (6-month-old). n = 5. **b** Representative bar graphs of elevated IgM autoantibodies in *Gcc2^+/+*, *Gcc2^−/−* and *Gcc2^−/−Sting^−/−* mouse serum (6-month-old). *n* = 5. Data are shown as mean ± s.e.m. *P* values are determined by one-way ANOVA, ns, not significant.

relevant examples of cGAS-STING activation also exist where cellular organelles appear to be normal and the pathogenic ligand was not immediately clear[6,11,27–30]. In addition, *cGas^−/−* cells or mice are more susceptible to a wide range of viral infections, many of which do not produce a classical 'trigger' of this pathway[2,31–33].

Can the cGAS-STING signaling be activated without incurring pathogenic DNA exposure? We think the answer is yes and relies on two

key elements. The first element is the understanding of homeostatic cGAS-STING signaling driven by non-pathogenic self-DNA in healthy wild-type cells, which supports basal level of ISG expression. This has been demonstrated before[2,20] and we further confirmed in this study. The second element is STING Golgi-exit that includes recycling back to the ER as well as forward post-Golgi trafficking to lysosomes. Impeding lysosomal degradation of STING can boost IFN signaling[10,11], although

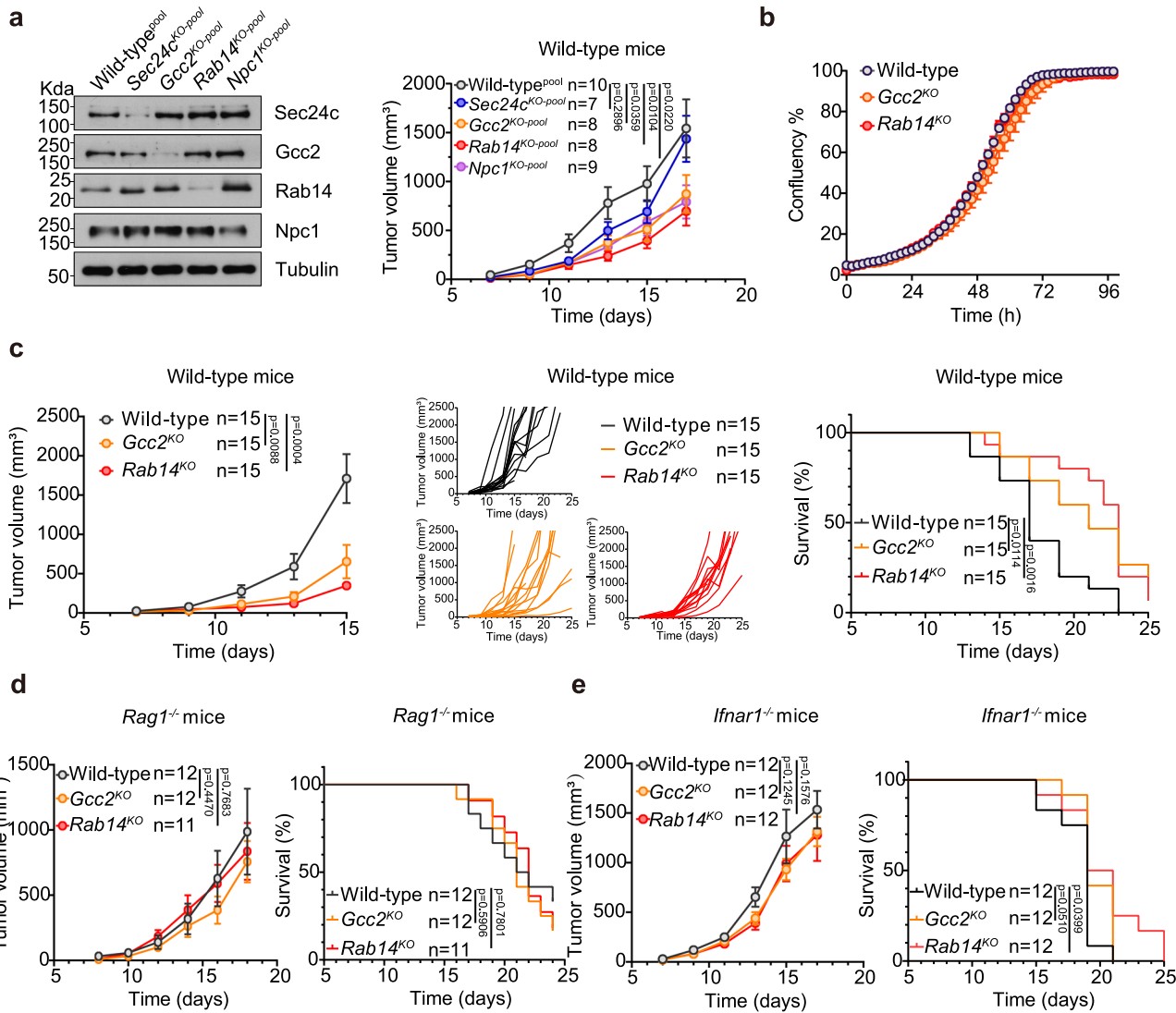

**Fig. 8 | Gcc2-KO and Rab14-KO induce anti-tumor immunity. a** B16 tumor growth in wild-type mice. Left, Western blot analysis of CRISPR/Cas9 knockout pooled B16 cells. Right, tumor volumes of wild-type^pool, *Sec24^KO-pool*, *Gcc2^KO-pool*, *Rab14^KO-pool*, or *Npc1^KO-pool* B16 melanoma cells after subcutaneous injection in wild-type mice. **b** Wild-type, *Gcc2^KO* and *Rab14^KO* B16 cell proliferation analysis by IncuCyte. Representative single-cell CRISPR/Cas9 knockout clones are shown. **c** Wild-type, *Gcc2^KO* and *Rab14^KO* B16 tumor growth in wild-type mice. Left, mean tumor volume. Middle, individual mouse tumor growth curve. Right, Kaplan–Meier survival curve. **d, e** Wild-type, *Gcc2^KO* and *Rab14^KO* B16 tumor growth in *Rag1^−/−* mice (**d**) and *Ifnar1^−/−* mice (**e**). Left, mean tumor volume, Right, Kaplan–Meier survival curve. Data in **b**, **c** are representative of two independent experiments and in **d**, **e** are pooled from two independent experiments. Data (**a–e**) are shown as mean ± s.e.m. *P* values are determined by two-way ANOVA in (**a–e**). ns, not significant. Mantel-Cox tests were used for survival studies.

how STING moves from the Golgi to the lysosome is not well understood. A major advance of this study is the discovery of post-Golgi regulators of STING trafficking. The majority of known STING cofactors act in the ER-to-Golgi phase, and very few cofactors are known in the Golgi-to-lysosome phase[34]. We found here that STING Golgi-exit is regulated by GCC2-RAB14, which is essential for STING degradation by the lysosome and attenuation of IFN signaling. Interestingly, STING-mediated NF-kB is not affected by post-Golgi interruption, revealing that different STING activities may rely on different segments of trafficking. The precise cell biology mechanism of GCC2-RAB14-STING trafficking requires further study. Nonetheless, *Gcc2^−/−* mice develop STING-dependent serologic autoimmunity, demonstrating the physiological importance of proper STING post-Golgi trafficking in vivo.

In terms of anti-tumor activity, *Gcc2^KO* and *Rab14^KO* can induce mechanistically similar anti-tumor immunity as STING agonists (T cell-dependent, IFN-dependent). We can consider STING agonists as 'fast and furious' at inducing anti-tumor immunity, which is potent but

often leads to toxicity. Post-Golgi trafficking interruption could be an alternative 'slow and steady' mode of STING activation, which may lead to more favorable outcomes as a cancer immunotherapy.

The *Gcc2*-deficiency mechanism is similar to *COPA*-deficiency. *COPA*-deficiency impairs retrograde COPI vesicle trafficking, leaving STING to accumulate on the Golgi and drive IFN signaling and tissue pathology in mice[6,7,27,30]. *Gcc2^−/−* impairs STING Golgi-exit as well as further trafficking to the lysosomes for degradation. The overall phenotype in *Gcc2*-deficiency is less severe compared to *Copa*-deficiency. For example, fewer STING protein molecules accumulate on the Golgi (by microscopy) in *Gcc2*-deficient cells than *Copa*-deficient cells; this correlates with in vivo phenotypes – *Gcc2*-deficient mice only develop moderate serological autoimmunity with no tissue pathology or immune cell dysregulation, whereas *Copa*-deficient mice have much more severe disease[7]. Importantly, both *Gcc2*-deficient and *Copa*-deficient mouse phenotypes are genetically STING-dependent. Further, both *Gcc2*-deficient and *Copa*-deficient cells require cGAS for elevated

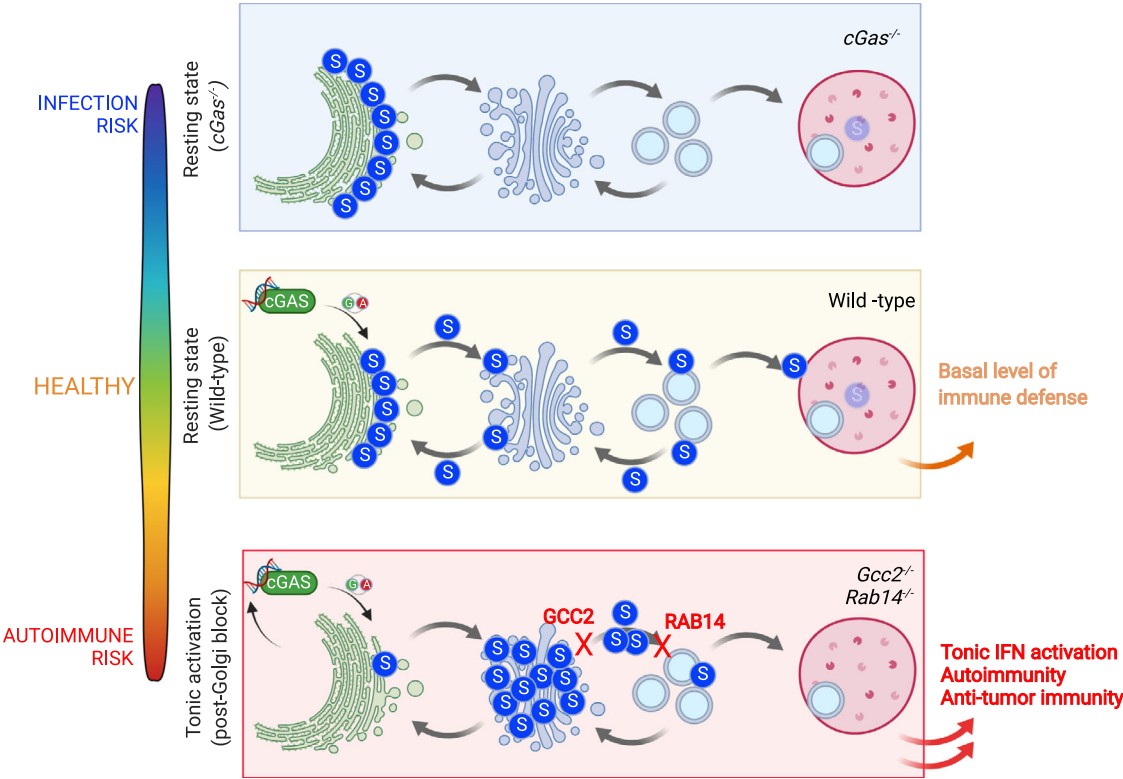

**Fig. 9 | An overall model.** Homeostatic cGAS activity drives STING trafficking that supports the basal level of immune defense and maintains a healthy state. *cGas*[-/-] cells and mice have lowered basal immune defense, leading to infection risk. Interruption of STING post-Golgi trafficking by *Gcc2*[-/-] and *Rab14*[-/-] leads to tonic IFN activation, increased autoimmunity and anti-tumor immunity.

ISG expression but do not appear to incur pathogenic self-DNA[30]. Both examples demonstrate that perturbing STING trafficking and retaining it on the Golgi can boost homeostatic cGAS-STING signaling and cause disease pathology. Moreover, different extents (or types) of STING trafficking interruption can lead to varying disease severity. It is also possible that at resting state STING recycling to the ER may be predominant over STING post-Golgi trafficking and degradation.

Another key insight from this study is that STING is actively moving through the secretory pathway at homeostasis rather than remaining stationary on the ER. Despite microscopy images of STING showing localization on the ER at the resting state, STING is much more dynamic. Many published studies as well as our data presented here show a small but detectable amount of STING signaling activity in resting wild-type cells, such as oligomerization, p-Sting, p-Tbk1, p-Stat1[35-37]. Compared to wild-type, *Sting*[-/-] mice and cells also show a lowered state of immune defense as measured by reduced basal expression of IFN and ISGs and increased susceptibility to infection[38-40]. *cGas*[-/-] also lowers the immune state whereas interrupting post-Golgi STING trafficking elevates the immune state. We thus propose a "basal-flux" mechanism of cGAS-STING signaling that operates at homeostasis in a healthy cell to set the basal state of immune defense. This is analogous to autophagic flux that is also operational at homeostasis to maintain cellular content turnover, and both over-activation and blockade of autophagic flux could lead to a pathogenic state. We believe that GCC2 and other post-Golgi trafficking cofactors uncovered in this study represent a growing list of previously unappreciated vesicle trafficking modulators that can tune cellular immune state through STING.

## Methods

### Mice and cells
Animal work was approved by the Institutional Animal Care and Use Committee at University of Texas Southwestern Medical Center. Wild-type (C57Bl/6J), *Rag1*[-/-], *Ifnar1*[-/-] mice were purchased from the Jackson Laboratory (#030097, #002216, #028288). *Gcc2*[+/-] mice (C57BL/6J-Gcc2[em1Cya], #S-KO-13297) and Rab14[+/-] (C57BL/6J-Rab14[em1Cya], #S-KO-13297) were purchased from Cyagen. All mice were housed in pathogen-free barrier facilities. Primary MEFs were isolated and cultured as described before[41]. B16F10 and HEK293T cells were obtained from ATCC. Cells were cultured in DMEM with 10% (vol/vol) FBS, 10 mM HEPES, 2 mM L-glutamine, and 1 mM sodium pyruvate with the addition of 100 U/mL penicillin and 100 mg/mL streptomycin, at 37 °C with 5% $CO_2$. Mycoplasma tests were conducted monthly and confirmed to be negative.

### Reagents and antibodies
Herring testis DNA (HT-DNA, D6898; Sigma) was used as dsDNA for stimulation of the cGAS-STING pathway. Synthetic Poly (I:C) HMW (tlrl-pic; InvivoGen) was used as dsRNA for stimulation of the RIG-I/MDA5-MAVS pathway. DMXAA (tlrl-dmx; Invitrogen) was used as a cell-permeable STING agonist. Primary antibodies used for Western blot included Sting (13647; CST, 1:1000), phospho-Sting (72971; CST, 1:1000), Tbk1 (3504; CST, 1:1000), phospho-Tbk1 (5483; CST, 1:1000), Irf3 (4302; CST, 1:1000), p65 (8242; CST, 1:1000), phospho-p65 (3033; CST, 1:1000), LC3B (MAB85582; R&D, 1:5000), Ifnar1 (ab124764; Abcam, 1:1000), Tubulin (T5168; Sigma, 1:20,000), Rab14 (15662-1-AP; Proteintech), cGAS (83623; CST 1:1000), Mavs (4983; CST, 1:1000), Sec24c (ab241336; Abcam, 1:1000), Npc1 (ab108921; Abcam, 1:1000), p-ATM (05-740; Sigma, 1:5000), FLAG (F1804; Sigma, 1:5000), FLAG (14793; CST, 1:5000), MYC (2276; CST, 1:5000), HA (3724; CST, 1:5000), Gcc2 antibody was kindly provided by Suzanne Pfeiffer (Stanford University). Primary antibodies used for microscopy imaging included Sting (19851-1-AP; Proteintech, 1:200), GM130 (610822; BD, 1:200), TGN38 (AHP499G; Bio-Rad, 1:200), Hsp60 (sc-13115; Santa Cruz, 1:200), PDI (ab2792; Abcam, 1:200), phospho-Tbk1 (5483; CST, 1:100), phospho-Irf3 (29047; CST, 1:200), HA (3724; CST, 1:1000).

Secondary antibody used included goat anti–rabbit IgG-HRP conjugate (1706515; Bio-Rad, 1:3000), goat anti–mouse IgG-HRP conjugate (1706516; Bio-Rad, 1:3000), TidyBlot Western Blot Detection Reagent:HRP (STAR209PA; Bio-Rad, 1:200), donkey anti-Rabbit IgG Alexa Fluor Plus 488 (#A-32790; ThermoFisher, 1:1000), donkey anti-Sheep IgG Alexa Fluor 594 (#A-11016; ThermoFisher, 1:1000), and donkey anti-Mouse IgG Alexa Fluor 647 (#A-31571; Thermo-Fisher, 1:1000).

## siRNA transfection, RNA isolation

Predesigned siRNA oligomers were purchased from Sigma-aldrich and dissolved in water to 20 μM. Reverse transfection was conducted by using Lipofectamine RNAiMAX reagent (13778150; ThermoFisher) with siRNA in Opti-MEM media (51985034; ThermoFisher). siRNAs used in this study are: simSec24c#2: SASI_Mm02_00344292, simGcc2#1: SASI_Mm02_00332069, sihGCC2#1 SASI_Hs01_00230098, sihGCC2#2 SASI_Hs02_00370060, sihGCC2#3 SASI_Hs01_00230101. MISSION siRNA Universal Negative Control #1 SIC001 was used as the siRNA control. Pooled siRNAs for secondary screen were purchased from Sigma-Aldrich. Total RNA was extracted with TRI reagent according to the manufacturer's protocol (T9424; Sigma-Aldrich). cDNA was generated using iScript Reverse Transcription Supermix kit (1708840; Bio-Rad). Quantitative RT–PCR (qPCR) was performed using iTaq Universal SYBR Green Supermix (1725120; Bio-Rad),CFX Connect Real-Time PCR Detection System (1855201;Bio-Rad) and Bio-Rad CFX Maestro software (12013758; Bio-Rad). We typically isolate 10 ng to 1 ug RNA from cells or tissues to generate cDNA, then use a small fraction of that cDNA for qPCR depending on how many target gene expression we need to measure. Primer sequences are provided in Source Data.

## CRISPR-Cas9, lentiviral, and retroviral transduction

LentiCRISPRV2 plasmids were used to generate knockout cell lines. Target guide sequences are Gcc2 #1:TTGGATTTCGGGGTCCCGGA #2:CTCCGGGACCCCGAAATCCA; Rab14#1: AATTCAACACCAATTGTGT, #2:GCTGATTGTCCTCACACAAT; cGAS#1: GCGGACGGCTTCTTAGCGCG, #2:AAAGGGGGGCTCGATCGCGG; Sting#1: CAGTAGTCCAAGTTCGTGCG, #2: GTCCAAGTTCGTGCGAGGCT; Npc1#1: GGACGGCTATGACTTAGTGC, #2: GGTGCGCACCACCCGGCCCT; Sec24c: TCGAGTTATGGTGGGCAACC, #2: AACTCGACTGATGATACCCT. Lentiviruses carrying the sgRNA and Cas9 were generated in HEK293T with the packaging plasmid psPAX2 and the envelope plasmid pMD2.G. Wild-type cGAS, E225A/D227A cGAS, K407A/K411A cGAS, wild-type STING, R238A/Y240A STING, S366A STING were cloned to a retroviral pMRX-IRES-Bsr vector (a gift from S. Akira) using HiFi DNA assembly (E2621S; NEB). Retroviruses were packaged in HEK293T cells with the packaging plasmid pGag-Pol and the envelope plasmid pVSV-G. Cells were selected with antibiotics after transduction for several days. Single cell clones were selected and verified by western blot for CRISPR-Cas9 knockout cell lines.

## Co-immunoprecipitation and western blot

Wild-type HEK293T cells were transfected with indicated plasmids, harvested 24 h later, washed once with PBS buffer, lysed in IP lysis buffer (20 mM Tris–HCl, pH 7.4, 0.5% Nonidet-P40, 150 mM NaCl and 1× protease inhibitor mixture) and centrifuged at 20,000 g for 20 min at 4 °C. In all, 5% of the supernatant was saved as input. In total, 25 μL of the Protein G dynabeads (10004D; ThermoFisher) were incubated with primary antibody for 2 h with rotation at room temperature. Equal amounts of protein (based on BCA assay) were incubated with antibody-bound beads with rotation overnight at 4 °C. Beads were washed three times with IP buffer. IP complex was eluted in 1X Western blot sample buffer (EC887; National Diagnostics, 5X) and boiled at 95 °C for 10 min. 5X Protein Loading Buffer from National Diagnostics contains 1.0 M TrisHCl (pH 8.5), 8% (w/v) lithium dodecyl sulfate, 40% (v/w) glycerol, 2 mM EDTA, 0.5 M DTT

and tracking dye in distilled/deionized water. 1X Western blot sample buffer was made from dilution of 5X Protein Loading buffer with water. For Western blot, equal amounts of protein were separated by SDS–PAGE, followed by transferring to nitrocellulose membrane. Membrane blots were blocked in 5% milk in TBST buffer for 30 min at room temperature, followed by incubation with primary antibodies in 3% BSA in TBST buffer at 4 °C overnight. After several washes, membrane blots were incubated with HRP-conjugated IgG secondary antibody for 1 h at room temperature. Then, membrane blots were developed with SuperSignal West Pico Chemiluminescent Substrate (34580; ThermoFisher) and blue autoradiography film (BDB810; Dot Scientific) The films were scanned using LIDE 300 scanner (2995C002; Canon).

## Endogenous cGAS immunoprecipitation and DNA analysis

Wild-type MEFs were washed once with PBS buffer, lysed in IP lysis buffer (20 mM Tris–HCl, pH 7.4, 0.5% Nonidet-P40, 150 mM NaCl and 1× protease inhibitor mixture) and centrifuged at 20,000 × g for 20 min at 4 °C. cGAS antibody from different vendors was incubated with 500 μg of cell lysate from wild-type MEFs with rotation at 4 °C overnight. Next day, 50 μL of the Protein G dynabeads (10004D; ThermoFisher) was incubated with the lysate and antibody mix for 1 h with rotation at 4 °C. cGAS antibodies used included cGas (ab252416; Abcam), cGas (31659; CST), cGas (ZRB1406; Sigma). Beads were washed three times with IP lysis buffer. 40% of the beads were saved for western blot and the rest used for DNA isolation. For Western blot, IP complex was eluted in 1X Western blot sample buffer and boiled at 95 °C for 10 min. For DNA isolation, beads were pelleted, resuspend in 100 μL of elution buffer (1% SDS, 100 mM NaHCO3) and boiled at 65 °C for 10 min. 0.5 μL of RNases (AM2288; Thermofisher) was added to the beads and beads were incubated at 37 °C for 30 min. Next, 5 μL of Proteinase K (AM2548; Thermofisher) was added to the beads and beads were incubated at 55 °C for 30 min and then 95 °C for 10 min. Finally, the IP DNA was isolated using QIAamp DNA mini kit (56304; Qiagen) and diluted 1:10 before qPCR analysis. Primer sequences are provided in Source Data.

## Inhibition of lysosome and proteasome

Wild-type MEFs were seeded in 6-well plates overnight. Next day, cells were pre-treated with media supplemented with DMSO (0.1%), 50 nM lysosome inhibitor Bafilomycin A1 (tlrl-baf1; Invivogen) or 10 μM proteasome inhibitor MG132 (M8699; Sigma) for 1 h. Then cells were transfected with 1 μg/mL HTDNA for 1 h before protein isolation and Western blot analysis of STING pathway activation.

## Fluorescent confocal microscopy

Cells were cultured on glass coverslips (for fixed-cell) or on glass bottom dishes (for live-cell) in Complete Medium. For fixed-cell microscopy, cells were fixed with ice-cold 100% Methanol for 10 min and blocked with 10% Power Block (HK0855K; Biogenex Laboratories). Coverslips were incubated with primary antibody overnight at 4 °C followed by incubation with fluorescence-conjugated secondary antibody for 1 h at room temperature. Coverslips were counterstained with 1 μg/mL DAPI solution (D9542; Sigma) for 5 min and rinsed with PBS-T/PBS for 5 min. Then coverslips were mounted on slides with Prolong Glass Antifade Mountant (P36982; ThermoFisher, no DAPI). Slides were imaged with Zeiss LSM 880 confocal microscope. Live-cell microscopy were imaged with Andor spinning-disc confocal microscope with environmental $CO_2$ and temperature controls. Image deconvolution was conducted using AutoQuant X software (Media Cybernetics), then analyzed using Fiji and Zen Blue (Zeiss).

## Infection

Cells were infected with HSV-1-GFP (moi = 1, determined by plaque assay) while occasionally shaking for 1 h and then washed and cultured

in fresh medium. After 24 h, supernatant was collected for plaque assay and mouse IFN-β ELISA. Cells were washed and collected for flow cytometry.

## Generation of Bone marrow-derived macrophages

Bone marrow was isolated from the mouse tibia and femur following aseptic technique. 1X RBC lysis buffer (420302; Biolegend) was added to the bone marrow for 5 min at room temperature. The lysis reaction was neutralized by 4-fold volume of DMEM with 10% FBS. Bone marrow cells were pelleted and plated in 10 cm dish in DMEM with 10% FBS/20% L929 cells-conditioned media for differentiation into bone-marrow derived macrophages (BMDMs). 10 mL of fresh DMEM with 10% FBS/20% L929 cells-conditioned media was added to the dish at day 3 and 5. Differentiated BMDMs were collected at day 6 for analysis.

## Flow cytometry

Peritoneal cavity cells were isolated by collecting 8 mL of peritoneum lavage fluids from the mouse. Cells were counted and equal number of cells were stained with TruStain FcX antibody (101320; Biolegend), live-dead Zombie Aqua dye (423102; Biolegend), APC anti-mouse CD19 antibody (152410; Biolegend), and BV421 anti mouse CD43 antibody (562958; BD) for the identification of peritoneal B cells. Mouse spleens were crushed and grinded on a 70 μM cell strainer (352350; Falcon) with a 5 mL syringe plunger. In total, 10 mL of RPMI with 10% FBS was added onto the cell strainer. Cells were pelleted and resuspend in 2 mL of 1X RBC lysis buffer (420302; Biolegend) for 5 min at RT. The lysis reaction was neutralized by 4-fold volume of DMEM with 10% FBS. Cells were counted and equal number of cells were stained with TruStain FcX antibody (101320; Biolegend), live-dead Zombie Aqua dye (423102; Biolegend). The following flow antibodies were used: PE anti-mouse CD3 antibody (100206; Biolegend), APC/Cy7 anti-mouse CD4 antibody (100414; Biolegend), FITC anti-mouse CD8 (100706; Biolegend), Pacific Blue anti-mouse CD44 (103020; Biolegend), PE/Cy7 anti-mouse CD62L (104418; Biolegend), PerCP/Cy5.5 anti-mouse CD19 (152406; Biolegend), FITC anti-mouse CD45R/B220 (103206; Biolegend), PE/Cy7 anti-mouse IgD (405720; Biolegend), and PE anti-mouse CD138 (142504; Biolegend). Cell surface Ifnar1 was examined by staining the cells with Biotin anti-mouse IFNAR1 (127306; Biolegend) and APC/Cy7 Streptavidin (405208; Biolegend) afterwards. The samples were acquired in a BD calibur flow cytometer using BD FACStation or in a Beckman flow cytometer using CytoExpert software.

## ELISA

Mouse IFN-β ELISA was performed as previously reported[41]. Cell-free supernatant was analyzed for the presence of mouse IFN-β by ELISA using paired antibodies (519202, 5081055; Biolegend). cGAMP ELISA (501700; Cayman Chemical) was performed according to manufacturer's instructions. The absorbance of the ELISA plate was measured by a microplate reader (Synergy HT; BioTek) using Gen 5 software (Gen5; BioTek).

## Autoantibody and cytokine array

Mouse serum was isolated by centrifuging clotted blood at 10,000×g for 10 min. Serum was immediately frozen at −80 °C until further analysis. Mouse serum was subjected to either Bio-Plex Pro Mouse Cytokine 23-plex assay (M60009RDPD; Bio-Rad) for serum cytokine analysis or autoantigen Microarray Super Panel I (UTSW) for autoantibody analysis.

## Temperature shift study

Before the temperature shifts, cells were cultured in DMEM with 10% (vol/vol) FBS, 10 mM HEPES, 2 mM L-glutamine, and 1 mM sodium pyruvate with the addition of 100 U/mL penicillin and 100 mg/mL streptomycin, at 37 °C with 5% CO2. Cell culture medium were replaced with either pre-warmed (37 °C) or pre-chilled (20 °C) CO2-independent medium (18045088; ThermoFisher) with 10% (vol/vol) heat-inactivated FBS, 10 mM HEPES, 2 mM L-glutamine, and 1 mM sodium pyruvate with the addition of 100 U/mL penicillin and 100 mg/mL streptomycin. Stimulation was conducted at either 37 °C or 20 °C by transfection of dsDNA for the indicated times.

## Mitochondrial DNA depletion

Cells were maintained in DMEM with 10% (vol/vol) FBS, 1 mM sodium pyruvate and 100 ug/mL uridine with or without addition of 150 ng/mL ethidium bromide for 6 days. Total DNA was isolated using QIAamp DNA mini kit (56304; Qiagen). Mitochondrial DLOOP2 gene was quantified using qPCR and normalized to genomic HPRT gene. Cells were replated overnight and treated with 500 nM ABT-737 (S1002; SelleckChem) and 20 μM Q-VD-Oph (S7311; SelleckChem) for the induction of *Ifnb1*. Total RNA was isolated to profile ISG expression by qPCR after reverse transcription. Primer sequences are provided in Source Data.

## Cytoplasmic DNA content profiling

Cytoplasmic DNA content profiling was performed as previously described[42]. Cells were resuspended in 500 μL extraction buffer (150 Mm NaCl, 50 Mm HEPES Ph = 7.4, and 20 μg/mL digitonin). In all, 100 μL was set aside as the whole cell fraction. The rest (400 μL) was incubated on ice for 10 min and then centrifuged at 1000 × g for 3 min. The supernatant was further subject to a high speed centrifugation at 17,000 × g for 10 min and then transferred to a new tube as the cytosolic fraction. DNA was isolated from both the whole cell fraction and cytosolic fraction using the QIAamp DNA mini kit (56304; Qiagen). Both fractions were eluted in 50 μL water and diluted 1:10 before qPCR analysis. DNA isolated from whole cell fraction was used as the input. DNA abundance from the cytosolic fraction was normalized to the abundance in the whole cell fraction using $2^{-(\Delta\Delta CT)}$ methods. Primers are provided in the source data.

## Tumor model

A total of $5 \times 10^5$ B16 or derivative knockout cells were subcutaneously injected to the flank of mice. Tumor size was measured every other day using a caliper and calculated as length × width × height/2. All protocols were approved and in compliance with UT Southwestern Institutional Animal Care and Use (IACUC).

## Statistical analysis

Statistical tests were noted in figure legends. All data were shown as means ± s.e.m, and all analyses were performed using Prism 9 software (Graphpad). Schematic graphs were created with Biorender.com.

## Reporting summary

Further information on research design is available in the Nature Research Reporting Summary linked to this article.

## Data availability

Source data for all figures are provided with the paper. Source data are provided with this paper.

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

## Acknowledgements

We thank Yang-Xin Fu and members of the Fu lab for advice on tumor experiments, Robert Hammer and Mylinh Nguyen at the UTSW Transgenic Technology Center, Kate Phelps and Abhijit Bugde at the UT Southwestern Live Cell Imaging Facility, a Shared Resource of the Harold C. Simmons Cancer Center, supported in part by NCI Cancer Center Support Grant 1P30 CA142543-01, NIH Shared Instrumentation Award 1S10 OD021684-01, members of the Yan lab for discussions. This work was also supported by National Institutes of Health (AI151708, NS117424, NS122825 to N.Y.), Cancer Prevention and Research Institute of Texas (CPRIT, RP180288, RP220242 to N.Y.), the Burroughs Wellcome Fund (N.Y.), UT Southwestern Immunology T32 training grant (5T32AI005284, to K.A. and D.J.).

## Author contributions

X.T. and N.Y. conceived and designed the study. X.T. performed most of the experiments. T.C. performed the primary screen and helped with siRNA validation. D.J, K.A., and N.D. helped with tumor experiments, microscopy, and mouse tissue analysis. K.Y. helped with the generation of retroviruses and FACS analysis of mouse immune cells. C.X. helped with mtDNA depletion and siRNA experiments. J.H. helped with mouse serum collection. X.T. and N.Y. wrote the paper with inputs from all coauthors.

## Competing interests

The authors declare no competing interests.
