## [Peer Review File · Nature Communications]

Interruption of post-Golgi STING trafficking activates tonic interferon signalingEditorial Note: This manuscript has been previously reviewed at another journal that is not operating a transparent peer review scheme. This document only contains reviewer comments and rebuttal letters for versions considered at *Nature Communications*.

REVIEWER COMMENTS

Reviewer #1 (Remarks to the Author):

I sincerely thank the authors for addressing all of my concerns and questions. It's also great to see the addition of the GCC/STING dKO experiments, which demonstrate the validity of these findings in vivo.

I strongly believe that the current version of the manuscript should be accepted at Nature Communications.

Reviewer #2 (Remarks to the Author):

Tu et al. provide a new version of their manuscript, revised after a submission at Nature Immunology, studying the post-Golgi trafficking of STING at baseline and upon stimulation with exogenous ligand, revealing a role for GCC2 and Rab14. The authors have addressed several important comments of the reviewers, including a detailed description of the GCC2 KO mouse phenotype, the addition of a trans Golgi marker, statistical details and a more comprehensive Methods section. Moreover, the authors have – in my view - more appropriately acknowledged previous literature on the existence of tonic cGAS-STING signalling (although, in some instances, the interpretation of their results still seems to gloss over this I would say).

Despite these improvements, there are certain key points which I consider still not to have been addressed:

-STING degradation: the data presented do not convincingly show that STING degradation by lysosomes is BLOCKED in GCC2 KO MEFs. STING is still lost upon HT-DNA stimulation in GCC2 KO cells (Fig 2d, line 106) and Rab14 KO cells (Fig 6d) and Bafilomycin A1 still has an effect on this degradation (Fig S2a). As mentioned before, a graph of quantification included in the figures for key WB would help support the claims of a difference. Summarising, I am not saying that STING degradation is unaffected (although the fold may be similar with quantification and a graph), but it is not 'blocked' - it is (might be) impaired. To me, this difference in terms is worth pointing out.

Of note, Bafilomycin A1 does not affect STING levels in WT cells suggesting there is no constant degradation of STING, and thus higher STING levels in GCC2 KO cells may not be due to lack of constant degradation.

I would like to point out again that STING mRNA tends to increase in RAB14 KO cells (Fig S7F).

What about in GCC2 KO cells?

Is there also altered degradation of STING in GCC2 knockdown THP1 and HeLa? This would be an interesting confirmatory piece of data, and could compensate for a very mildly affected IFN signalling in these cells (Fig S2).

-the link between GCC2 and STING: The effect of GCC2 KO on post-Golgi trafficking seems rather specific to STING since the secretory pathway appears unaffected (line 95). Yet the authors provide no explanation for this specificity, either experimentally or through discussion, which remains an

important gap in this study. The authors explain that the lack of interaction between STING and GCC2 is likely due to a rapid hand-off of STING by GCC2. However, interactions between GCC2 and Rab14 and between Rab14 and STING are detected. Can the authors comment this? Can the authors show that STING, GCC2 and Rab14 cofractionate in endosomes and/or TGN compartments by biochemical fractionation, at baseline or upon stimulation? Or by colocalization of STING, GCC2, and Rab14 with key markers of post-Golgi compartments as in Gui et al. (PMID: 30842662).

As support for this, what is the STING signalling phenotype of GCC2/Rab14 double KO cells?

Line 226: stating that GCC2 quickly hands off STING is misleading as no interaction is shown at all.

Overall, this seems like a missed opportunity to generate insight into some of the precise compartments where STING locates in the post Golgi part of the secretory pathway.

-As the authors now point out in clarifying some of their results, the phenotype in GCC2 KO cells or mice seems weaker than for COPA deficiency. This may suggest that STING recycling to the ER may be predominant over STING post-Golgi trafficking and degradation at baseline. This may be an important conclusion to include in the discussion.

Other comments:

-The title has become quite confusing and needs to be revised.

-I would be more careful with saying that the autophagy pathway is intact in GCC2 KO cells (Line 111). A lower LC3bII/LC3bI ratio seen in KO cells suggests altered autophagy, which could contribute to STING homeostasis signalling (PMID: 29496741).

-For the ABT/QVD treatment, my point was to show that the authors could detect an increase in cytosolic mtDNA levels (as it can be notoriously delicate to do so), as opposed to looking at IFN signalling, which is well established in MEFs. Indeed, they show that there is no change in cytosolic mtDNA levels between WT and GCC2 KO cells. ABT/QVD would act as a positive control to show that the authors are able to detect a change in cytosolic mtDNA levels by qPCR if present.

-To address a possible effect of the 20°C setting on DNA transfection (delayed signalling), the authors use DMXAA (non-transfected STING agonist) and polyIC (transfected non-STING agonist) as controls (Fig S5). However, it is clear that signalling upon DMXAA is also delayed (and sustained) at 20°C, whereas it does not appear so with PolyIC, suggesting that STING signalling more generally is slowed down at 20°C. This could be the reason for sustained signalling i.e. because of slower trafficking, rather than a “global arrest of post-Golgi trafficking”. Accordingly, I would replace “global arrest” with something like “slowed down trafficking”, for the 20°C experiment (Fig 3 and S5 and corresponding text paragraph).

-The fact that cGAS binds to mtDNA and that mtDNA depletion by EtBr does not change IFN signaling in GCC2 KO cells is contradictory. The authors should comment on this.

-I understand that the authors do not want to embark on a whole description of the ligands and location of cGAS basal signalling is in this paper. And, indeed, showing similar levels of cGAMP in WT and GCC2 KO cells is consistent with an increase in signalling at the level of STING rather than cGAS ligand generation (Fig 4i). In this case, although the authors have moderated some statements, the text still contains overly strong interpretation of their data on this topic, notably that they show for the first time the existence of basal cGAS signalling.

-Line 86-88: “We also observed an interesting ER-exit delay in Gcc2 knockdown cells, which may be a feedback mechanism when STING is accumulated on the Golgi and active retrograde trafficking transporting STING back to the ER” does not seem in the right location, rather linked to Fig 2g. Of

note, there is no delayed trafficking in what seems like a similar experiment in Fig S4C, which the authors may want to switch for Fig 2g.

Minor comments:

-There is still no detail about the amount of RNA and cDNA used for reverse transcription and qPCR, respectively, in the Methods.

-I understand that the authors do not want to pursue further the study of RAB14 KO accelerating ER-exit; however, for accuracy, I still think the 'ER-exit' bar should be put earlier for Rab14 KO cells, i.e. at 0.25h for KO cells in Fig 6e.

This bar is also not in the right place in Fig 3c for 20°C (rather at 4h), and in Fig 2g for GCC2 KO cells (rather at 0.75h).

-# 35, 31 and 36 are duplicated in references.

-Line 46: "damping" - "dampening"

-Line 84: We do not see that STING is accumulating on the ER or Golgi, rather in the cytoplasm overall.

-Line 85: There is no data to support the statement: "small amount of STING protein accumulation on the Golgi"

-Line 101: Add a figure reference for double KO cells.

-Fig S3d: How was the Golgi area determined for live imaging?

-Line 285: The second element is rather 'Golgi exit' than 'post-Golgi trafficking', since it includes recycling to the ER.

-Line 289: Add references.

-Line 317: Replace 'propose' by 'confirm'.

Reviewer #1 (Remarks to the Author):

I sincerely thank the authors for addressing all of my concerns and questions. It's also great to see the addition of the GCC/STING dKO experiments, which demonstrate the validity of these findings in vivo.

I strongly believe that the current version of the manuscript should be accepted at Nature Communications.

We thank R#1 for supporting publication of our study.

Reviewer #2 (Remarks to the Author):

Tu et al. provide a new version of their manuscript, revised after a submission at Nature Immunology, studying the post-Golgi trafficking of STING at baseline and upon stimulation with exogenous ligand, revealing a role for GCC2 and Rab14. The authors have addressed several important comments of the reviewers, including a detailed description of the GCC2 KO mouse phenotype, the addition of a trans Golgi marker, statistical details and a more comprehensive Methods section. Moreover, the authors have – in my view - more appropriately acknowledged previous literature on the existence of tonic cGAS-STING signalling (although, in some instances, the interpretation of their results still seems to gloss over this I would say).

Despite these improvements, there are certain key points which I consider still not to have been addressed:

-STING degradation: the data presented do not convincingly show that STING degradation by lysosomes is BLOCKED in GCC2 KO MEFs. STING is still lost upon HT-DNA stimulation in GCC2 KO cells (Fig 2d, line 106) and Rab14 KO cells (Fig 6d) and Bafilomycin A1 still has an effect on this degradation (Fig S2a). As mentioned before, a graph of quantification included in the figures for key WB would help support the claims of a difference. Summarising, I am not saying that STING degradation is unaffected (although the fold may be similar with quantification and a graph), but it is not 'blocked' - it is (might be) impaired. To me, this difference in terms is worth pointing out.

We changed 'blocked' to 'impaired' to better describe the STING degradation phenotype. In GCC2-KO or RAB14-KO cells, STING could still be degraded by proteasomes or spontaneous encounters with lysosomes. As for better visualization of the WB data, we now include quantification graphs next to key WBs (We also have quantification of all WBs in Source Data in table format):

- Figure 2d (WT vs Gcc2-KO), we included Sting/Tub graph of WB quantification.
- Figure 6d (WT vs Rab14-KO), we included Sting/Tub graph of WB quantification.

Of note, Bafilomycin A1 does not affect STING levels in WT cells suggesting there is no constant degradation of STING, and thus higher STING levels in GCC2 KO cells may not be due to lack of constant degradation. I would like to point out again that STING mRNA tends to increase in RAB14 KO cells (Fig S7F). What about in GCC2 KO cells?

We see a slight trend of increase of STING mRNA level in Rab14-KO and Gcc2-KO cells, 1.2-fold and 1.5-fold respectively (Fig. S7f, Fig. S1f). These are not statistically significant. Comparing STING mRNA or protein levels at the baseline is one of several lines of evidence for GCC2 or RAB14 mechanism of action on STING. DMXAA-induced STING degradation data in Figure 2d and 6d are much more convincing at showing impaired STING degradation in Gcc2-KO and Rab14-KO cells.

Is there also altered degradation of STING in GCC2 knockdown THP1 and HeLa? This would be an interesting confirmatory piece of data, and could compensate for a very mildly affected IFN signalling in these cells (Fig S2).

We did not pursue biochemical characterization of STING degradation in THP1 and HeLa cells because the effect size of IFN signaling change is small (although statistically significant). e.g. GCC2 knockdown increased IFNB1 mRNA expression by 50%. We do not think WB blots like we did in Figure 2d and 6d will be able to capture that difference. Nonetheless, these experiments are important confirmation of the GCC2 mechanism in

human cells. The small effective size in IFN signaling we observed is likely due to partial knockdown.

-the link between GCC2 and STING: The effect of GCC2 KO on post-Golgi trafficking seems rather specific to STING since the secretory pathway appears unaffected (line 95). Yet the authors provide no explanation for this specificity, either experimentally or through discussion, which remains an important gap in this study. The authors explain that the lack of interaction between STING and GCC2 is likely due to a rapid hand-off of STING by GCC2. However, interactions between GCC2 and Rab14 and between Rab14 and STING are detected. Can the authors comment this? Can the authors show that STING, GCC2 and Rab14 cofractionate in endosomes and/or TGN compartments by biochemical fractionation, at baseline or upon stimulation? Or by colocalization of STING, GCC2, and Rab14 with key markers of post-Golgi compartments as in Gui et al. (PMID: 30842662).

As support for this, what is the STING signalling phenotype of GCC2/Rab14 double KO cells?

Line 226: stating that GCC2 quickly hands off STING is misleading as no interaction is shown at all. Overall, this seems like a missed opportunity to generate insight into some of the precise compartments where STING locates in the post Golgi part of the secretory pathway.

We think the exact cell biology mechanism of GCC2/RAB14/STING hand-off will require an entire separate study. The key messages of the current study are: 1) GCC2 and RAB14 are essential post-Golgi trafficking regulators of STING, identified from a biochemical screen; and 2) immune signaling activation mechanism in Gcc2-KO is depending on basal-flux cGAS-STING signaling in vitro and in vivo. We did our best with the current cell biology dataset, and we certainly would like to learn more in the future, which would involve biochemical fractionation approaches, high-res microscopy approaches, and others to precisely define these events. These experimental systems will take substantial effort to establish in an immunology lab and we feel this exceeds the scope of the current study.

We removed the 'hand-off' model speculation in Results (line 226-228). We also added in discussion: "the precise cell biology mechanism requires further study" (line 290).

-As the authors now point out in clarifying some of their results, the phenotype in GCC2 KO cells or mice seems weaker than for COPA deficiency. This may suggest that STING recycling to the ER may be predominant over STING post-Golgi trafficking and degradation at baseline. This may be an important conclusion to include in the discussion.

We added this sentence in the discussion where we compare GCC2-KO to COPA-deficiency (line 308).

Other comments:

-The title has become quite confusing and needs to be revised.

We changed the title to "Tonic cGAS-STING activation through post-Golgi trafficking interruption". We are open to suggestions.

-I would be more careful with saying that the autophagy pathway is intact in GCC2 KO cells (Line 111). A lower LC3bII/LC3bI ratio seen in KO cells suggests altered autophagy, which could contribute to STING homeostasis signalling (PMID: 29496741).

We noted this in the revised text "NF- κ B was not affected, autophagy was slightly altered" (line 108).

-For the ABT/QVD treatment, my point was to show that the authors could detect an increase in cytosolic mtDNA levels (as it can be notoriously delicate to do so), as opposed to looking at IFN signalling, which is well established in MEFs. Indeed, they show that there is no change in cytosolic mtDNA levels between WT and GCC2 KO cells. ABT/QVD would act as a positive control to show that the authors are able to detect a change in cytosolic mtDNA levels by qPCR if present.

My lab has published on mtDNA activation of the cGAS-STING pathway before (Yang 2018 J. Exp. Med., Pokatayev 2020 Nat. Immunol.). We are confident that we can measure mtDNA level correctly.

-To address a possible effect of the 20°C setting on DNA transfection (delayed signalling), the authors use DMXAA (non-transfected STING agonist) and polyIC (transfected non-STING agonist) as controls (Fig S5). However, it is clear that signalling upon DMXAA is also delayed (and sustained) at 20°C, whereas it does not appear so with PolyIC, suggesting that STING signalling more generally is slowed down at 20°C. This could be the reason for sustained signalling i.e. because of slower trafficking, rather than a “global arrest of post-Golgi trafficking”. Accordingly, I would replace “global arrest” with something like “slowed down trafficking”, for the 20°C experiment (Fig 3 and S5 and corresponding text paragraph).

We changed “global arrest” to “slowdown trafficking” in maintext (line 145-160) and Fig. 3 legend.

-The fact that cGAS binds to mtDNA and that mtDNA depletion by EtBr does not change IFN signaling in GCC2 KO cells is contradictory. The authors should comment on this.

cGAS has no specificity for DNA substrate and binds to many sources of self-DNA in a healthy wild-type cell in immunoprecipitation experiments, including mtDNA, retroelement DNA and satellite DNA (shown in Figure S6a, also shown in Gentili et al Cell Reports 2019, pmid: 30811988). The EtBr experiment excludes mtDNA as the source of cGAS activation. pATM experiment excludes DNA damage. Therefore, it is mostly likely other form of nuclear DNA that is activating cGAS, and the most abundant cGAS substrate from the IP experiments is major satellite DNA in the nucleus.

-I understand that the authors do not want to embark on a whole description of the ligands and location of cGAS basal signalling is in this paper. And, indeed, showing similar levels of cGAMP in WT and GCC2 KO cells is consistent with an increase in signalling at the level of STING rather than cGAS ligand generation (Fig 4i). In this case, although the authors have moderated some statements, the text still contains overly strong interpretation of their data on this topic, notably that they show for the first time the existence of basal cGAS signalling.

We removed all indications of “first-time discovery of basal cGAS signaling”. We actually made it clear in Discussion: “This has been demonstrated before and we further confirmed in this study” (line 281). The key finding here is that basal cGAS-STING activity drives Gcc2-KO phenotype.

-Line 86-88: “We also observed an interesting ER-exit delay in Gcc2 knockdown cells, which may be a feedback mechanism when STING is accumulated on the Golgi and active retrograde trafficking transporting STING back to the ER” does not seem in the right location, rather linked to Fig 2g. Of note, there is no delayed trafficking in what seems like a similar experiment in Fig S4C, which the authors may want to switch for Fig 2g.

We move this statement to the Fig 2g as suggested (line 134).

Minor comments:

-There is still no detail about the amount of RNA and cDNA used for reverse transcription and qPCR, respectively, in the Methods.

We typically isolate 10 ng to 1 ug RNA from cells or tissues to generate cDNA, then use a small fraction of that cDNA for qPCR depending on how many target gene expression we need to measure. This is standard molecular biology that should be available in most labs. The amount of RNA needed for a successful RT-qPCR experiment is also dependent on reagents and instruments. We have added this to Methods (line 389-391).

-I understand that the authors do not want to pursue further the study of RAB14 KO accelerating ER-exit; however, for accuracy, I still think the ‘ER-exit’ bar should be put earlier for Rab14 KO cells, i.e. at 0.25h for KO cells in Fig 6e.

We modified this figure 6e as requested.

This bar is also not in the right place in Fig 3c for 20°C (rather at 4h), and in Fig 2g for GCC2 KO cells (rather

at 0.75h).

We modified this figure 3c and 2g as requested.

-# 35, 31 and 36 are duplicated in references.

Fixed.

-Line 46: “damping” - “dampening”

Fixed.

-Line 84: We do not see that STING is accumulating on the ER or Golgi, rather in the cytoplasm overall.

We meant after DNA stimulation. We fixed this long sentence to shorted sentences that make more clear statements (line 82-84).

-Line 85: There is no data to support the statement: “small amount of STING protein accumulation on the Golgi”

We removed this statement.

-Line 101: Add a figure reference for double KO cells.

Added.

-Fig S3d: How was the Golgi area determined for live imaging?

In the STING-GFP live cell experiment, GFP fluorescent was measured every 5 minutes by the Andor Spin-disk confocal microscope. ER-exit was estimated by increase of GFP signal and Golgi-exit was estimated by decrease of GFP signal. These are not based on organelle marker colocalization. The point we are trying to make here is that the STING protein curve and degradation kinetics from live cell imaging are very similar to those observed from fixed imaging studies (based on organelle marker colocalization).

We added this clarification to Fig S3d legend. We also changed solid lines to dotted lines in the figure S3d.

-Line 285: The second element is rather ‘Golgi exit’ than ‘post-Golgi trafficking’, since it includes recycling to the ER.

Fixed.

-Line 289: Add references.

Added.

-Line 317: Replace ‘propose’ by ‘confirm’.

Fixed.

REVIEWER COMMENTS

Reviewer #2 (Remarks to the Author):

Thank you and well done.